# Early Release Science of the exoplanet WASP-39b with JWST NIRSpec G395H

Lili Alderson[1 ✉], Hannah R. Wakeford[1 ✉], Munazza K. Alam[2], Natasha E. Batalha[3], Joshua D. Lothringer[4], Jea Adams Redai[5], Saugata Barat[6], Jonathan Brande[7], Mario Damiano[8], Tansu Daylan[9], Néstor Espinoza[10,11], Laura Flagg[12,13], Jayesh M. Goyal[14], David Grant[1], Renyu Hu[8,15], Julie Inglis[15], Elspeth K. H. Lee[16], Thomas Mikal-Evans[17], Lakeisha Ramos-Rosado[11], Pierre-Alexis Roy[18,19], Nicole L. Wallack[2,15], Natalie M. Batalha[20], Jacob L. Bean[21], Björn Benneke[18,19], Zachory K. Berta-Thompson[22], Aarynn L. Carter[20], Quentin Changeat[23,24], Knicole D. Colón[25], Ian J. M. Crossfield[7], Jean-Michel Désert[6], Daniel Foreman-Mackey[26], Neale P. Gibson[27], Laura Kreidberg[17], Michael R. Line[28], Mercedes López-Morales[5], Karan Molaverdikhani[29,30], Sarah E. Moran[31], Giuseppe Morello[32,33,34], Julianne I. Moses[35], Sagnick Mukherjee[20], Everett Schlawin[36], David K. Sing[11,37], Kevin B. Stevenson[38], Jake Taylor[18,19,39], Keshav Aggarwal[40], Eva-Maria Ahrer[41,42], Natalie H. Allen[11], Joanna K. Barstow[43], Taylor J. Bell[44], Jasmina Blecic[45,46], Sarah L. Casewell[47], Katy L. Chubb[48], Nicolas Crouzet[49], Patricio E. Cubillos[50,51], Leen Decin[52], Adina D. Feinstein[21], Joanthan J. Fortney[20], Joseph Harrington[53,54], Kevin Heng[42,55], Nicolas Iro[56], Eliza M.-R. Kempton[57], James Kirk[5,58], Heather A. Knutson[15], Jessica Krick[59], Jérémy Leconte[60], Monika Lendl[61], Ryan J. MacDonald[12,13,62], Luigi Mancini[17,63,64], Megan Mansfield[36], Erin M. May[38], Nathan J. Mayne[65], Yamila Miguel[49,66], Nikolay K. Nikolov[10], Kazumasa Ohno[20], Enric Palle[32], Vivien Parmentier[39,67], Dominique J. M. Petit dit de la Roche[61], Caroline Piaulet[18,19], Diana Powell[5], Benjamin V. Rackham[68,69], Seth Redfield[70,71], Laura K. Rogers[72], Zafar Rustamkulov[11], Xianyu Tan[39], P. Tremblin[73], Shang-Min Tsai[39], Jake D. Turner[12,13], Miguel de Val-Borro[74], Olivia Venot[75], Luis Welbanks[28], Peter J. Wheatley[41,42] & Xi Zhang[76]

Measuring the abundances of carbon and oxygen in exoplanet atmospheres is considered a crucial avenue for unlocking the formation and evolution of exoplanetary systems[1,2]. Access to the chemical inventory of an exoplanet requires high-precision observations, often inferred from individual molecular detections with low-resolution space-based[3–5] and high-resolution ground-based[6–8] facilities. Here we report the medium-resolution ($R \approx 600$) transmission spectrum of an exoplanet atmosphere between 3 and 5 μm covering several absorption features for the Saturn-mass exoplanet WASP-39b (ref. [9]), obtained with the Near Infrared Spectrograph (NIRSpec) G395H grating of JWST. Our observations achieve 1.46 times photon precision, providing an average transit depth uncertainty of 221 ppm per spectroscopic bin, and present minimal impacts from systematic effects. We detect significant absorption from $CO_2$ (28.5$\sigma$) and $H_2O$ (21.5$\sigma$), and identify $SO_2$ as the source of absorption at 4.1 μm (4.8$\sigma$). Best-fit atmospheric models range between 3 and 10 times solar metallicity, with sub-solar to solar C/O ratios. These results, including the detection of $SO_2$, underscore the importance of characterizing the chemistry in exoplanet atmospheres and showcase NIRSpec G395H as an excellent mode for time-series observations over this critical wavelength range[10].

We obtained a single-transit observation of WASP-39b using the NIRSpec[11,12] G395H grating on 30–31 July 2022 between 21:45 and 06:21 UTC using the Bright Object Time Series mode. WASP-39b is a hot ($T_{eq} = 1{,}120$ K), low-density giant planet with an extended atmosphere. Previous spectroscopic observations have shown prominent atmospheric absorption by Na, K and $H_2O$ (refs. [3,4,13–15]), with tentative evidence of $CO_2$ from infrared photometry[4]. Atmospheric models fitted to the spectrum have inferred metallicities (amount of heavy elements relative to the host star) from 0.003 to 300 times solar[3,15–20], which makes it difficult to ascertain the formation pathway of the planet[21,22]. The host, WASP-39, is a G8-type star that shows little photometric variability[23] and has nearly solar elemental abundance patterns[24]. The quiet host and extended planetary atmosphere make WASP-39b an ideal exoplanet for transmission spectroscopy[25]. The transmission spectrum of WASP-39b was observed as part of the JWST Transiting Exoplanet Community Director's Discretionary Early Release Science (JTEC ERS) Program[26,27]

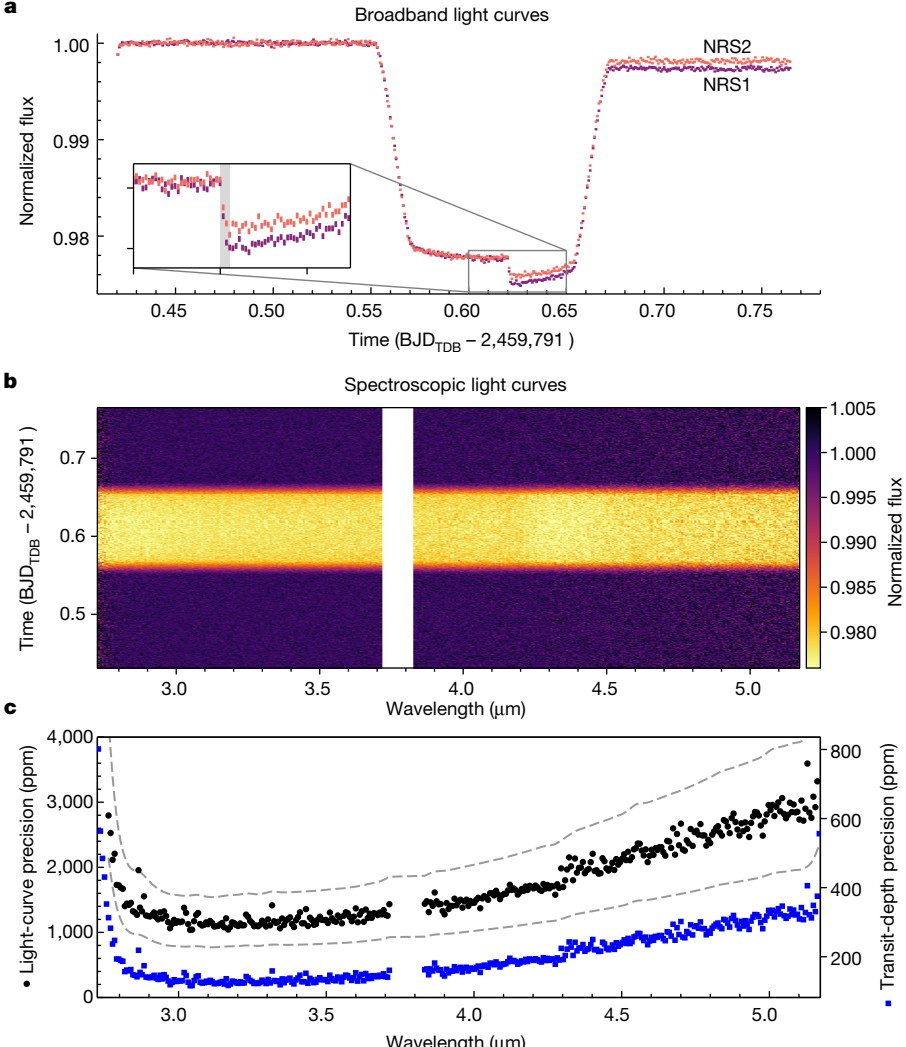

**Fig. 1 | Light-curve precisions achieved for WASP-39b with NIRSpec G395H.**
**a**, Raw, uncorrected broadband light curves from the NRS1 (purple) and NRS2
(red) detectors, demonstrating the lack of dominant systematic trends in the
light curves. The inset shows the drop in flux (grey-shaded region) caused by a
mirror-tilt event, resulting in a distinct change in flux between NRS1 and NRS2
after the tilt event (see Extended Data Figs. 2 and 3). **b**, Pixel intensity map of the
spectroscopic light curves after correction for the tilt event and further
instrument systematics. **c**, Light-curve precision obtained per spectroscopic
bin (black) compared with 1 and 2 times photon noise expectations (grey
dashed lines) and the measured precision on the transit depth (blue). The gap
between the two detectors (3.72–3.82 μm) is highlighted in the middle and
bottom plots. All data shown are from fitting pipeline 1 (see Methods).

(ERS-1366; principal investigators Natalie M. Batalha, Jacob L. Bean
and Kevin B. Stevenson), which uses four instrument configurations to
test their capabilities and provide lessons learned for the community.

The NIRSpec G395H data were recorded with the 1.6″ × 1.6″ fixed slit
aperture using the SUB2048 subarray and NRSRAPID readout pattern,
with spectra dispersed across both the NRS1 and NRS2 detectors. Over
the roughly 8-h duration of the observation, a total of 465 integrations
were taken, centred around the 2.8-h transit. We obtained 70 groups per
integration, resulting in an effective integration time of 63.14 s. During
the observation, the telescope experienced a 'tilt event', a spontaneous
and abrupt change in the position of one or more mirror segments,
causing changes in the point spread function (PSF) and hence jumps in
flux[28]. The tilt event occurred mid-transit, affecting approximately three
integrations and resulted in a noticeable step in the flux time series,
the size of which is dependent on wavelength (Fig. 1 and Methods). The
tilt event also affects the PSF, with the full width at half maximum
(FWHM) of the spectral trace showing a step-function-like shape (see
Extended Data Figs. 2 and 3).

We produced several reductions of the observations using inde-
pendent analysis pipelines (see Methods). For each reduction, we

created broadband and spectroscopic light curves in the ranges 2.725–
3.716 μm for NRS1 and 3.829–5.172 μm for NRS2 using 10-pixel-wide
bins (≈0.007 μm, median resolution $R \approx 600$), excluding the detector
gap between 3.717–3.823 μm. The light curves show a settling ramp
during the first ten integrations (≈631.4 s), with a linear slope across the
entire observation for NRS1. We otherwise see no substantial systematic
trends and achieve improvements in precision from raw uncorrected to
fitted broadband light curves of 1.63 to 1.03 times photon noise for NRS1
and 1.95 to 1.31 times for NRS2. The flux jump caused by the mirror-tilt
event could be corrected by detrending against the spectral trace $x$ and
$y$ positional shifts, normalizing the light curves or fitting the light curves
with a step function (see Methods). We produced several fits from each
set of light curves, resulting in a total of 11 independently measured
transmission spectra. Figure 1 demonstrates that our spectroscopic
light curves achieve precisions close to photon noise, with a median
precision of 1.46 times photon noise across the full wavelength range
(see Extended Data Fig. 4).

We show transmission spectra from several combinations of inde-
pendent reductions and light-curve-fitting routines in Fig. 2, along
with the weighted average of all 11 transmission spectra with the

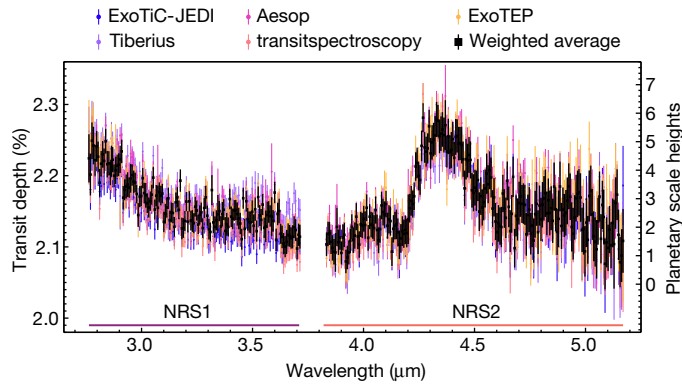

**Fig. 2 | WASP-39b transmission spectra measured at 10-pixel resolution (≈7-nm-wide bins, $R ≈ 600$) using several fitting pipelines.** We show the resultant spectra from five out of 11 independent fitting pipelines, which used distinct analysis methods to demonstrate the robust structure of the spectrum (see Methods for details on each fitting pipeline and comparative statistics). The black points show the weighted-average transmission spectrum computed from the transit depth values in each bin weighted by $1/\sigma^2$, in which $\sigma$ is the uncertainty on the data point from each of the 11 fitting pipelines. The error bars were computed from the unweighted mean uncertainty in each bin (see Extended Data Fig. 5). All spectra show consistent broadband absorption short of 3.7 μm, around 4.1 μm and from 4.2 to 4.5 μm.

unweighted mean uncertainty produced by our analyses (see Methods). We find that using different combinations of reduction and fitting methods results in consistent transmission spectra (see Methods and Extended Data Fig. 5). Although we see some artefacts at the edges of the detectors (see Fig. 3, bottom panel) that may be caused by uncharacterized systematics, these only affect a small number of wavelength bins. Our resulting averaged NIRSpec G395H spectrum shows increased absorption towards bluer wavelengths short of 3.7 μm and a prominent absorption feature between 4.2 and 4.5 μm, along with a smaller-amplitude absorption feature at 4.1 μm and a narrow feature around 4.56 μm.

We compared the weighted-average G395H transmission spectrum to three grids of 1D radiative–convective–thermochemical equilibrium (RCTE) atmosphere models of WASP-39b (see Methods and Extended Data Table 2), containing a total of 10,308 model spectra. The best-fit models from each grid provide a reduced chi-square per data point ($\chi^2/N$) of 1.08–1.20 for our 344-data-point transmission spectrum (Fig. 3). The increased absorption at blue wavelengths across NRS1 is consistent with absorption from $H_2O$ (at 21.5$\sigma$; see Methods), whereas the large bump in absorption between 4.2 and 4.5 μm (ref. [29]) can be attributed to $CO_2$ (28.5$\sigma$). $H_2O$ and $CO_2$ are expected atmospheric constituents for near-solar atmospheric metallicities, with the $CO_2$ abundance increasing nonlinearly with higher metallicity[30]. The spectral feature at 4.56 μm (3.3$\sigma$) is unidentified at present but does not correlate with any obvious detector artefacts and is reproduced by several independent analyses. The absorption feature at 4.1 μm is also not seen in the RCTE model grids. After an exhaustive search for possible opacity sources (S.-M. Tsai et al., manuscript in preparation), described in the corresponding NIRSpec PRISM analysis[31], we interpret this feature as $SO_2$ (4.8$\sigma$), as it is the best candidate at this wavelength.

Although $SO_2$ would have volume mixing ratios (VMRs) of less than $10^{-10}$ throughout most of the observable atmosphere in thermochemical equilibrium, coupled photochemistry of $H_2S$ and $H_2O$ can produce $SO_2$ on giant exoplanets, with the resulting $SO_2$ mixing ratio expected to increase with increasing atmospheric metallicity[32–34]. We find that a VMR of approximately $10^{-6}$ of $SO_2$ is required to fit the spectral feature at 4.1 μm in the transmission spectrum of WASP-39b, consistent with lower-resolution NIRSpec PRISM observations of this planet[31] and previous photochemical modelling of super-solar metallicity giant exoplanets[34,35]. Figure 4 shows a breakdown of the contributing opacity sources for the lowest $\chi^2/N$ best-fit model (PICASO 3.0) with VMR = $10^{-5.6}$ injected $SO_2$. The inclusion of $SO_2$ in the models results in an improved $\chi^2/N$ and is detected at 4.8$\sigma$ (see Methods), confirming its presence in the atmosphere of WASP-39b.

We also look for evidence of $CH_4$, CO, $H_2S$ and OCS (carbonyl sulfide) because their near-solar chemical equilibrium abundances could result in a contribution to the spectrum. We see no evidence of $CH_4$ in our spectrum between 3.0 and 3.6 μm (ref. [23]), which is indicative of C/O < 1

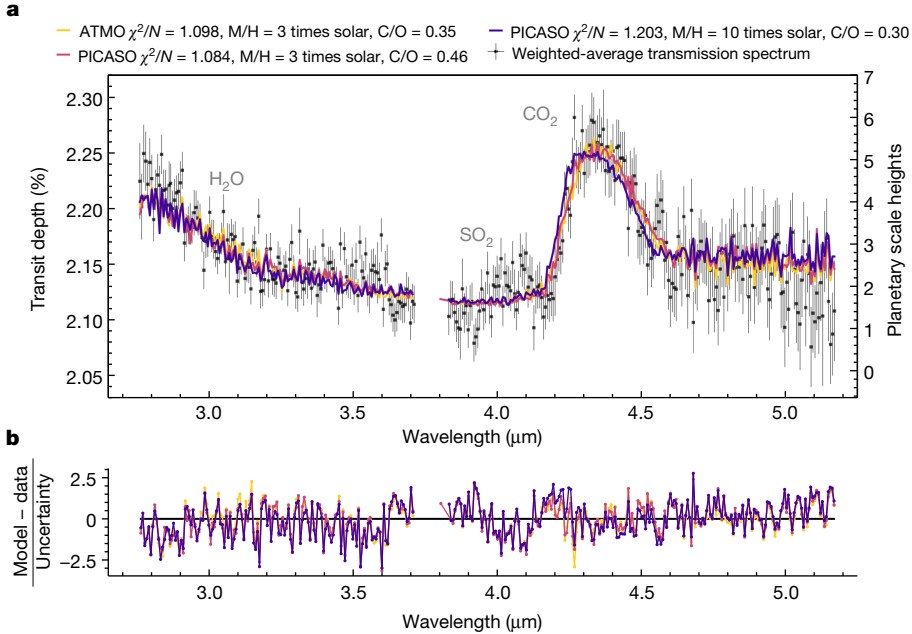

**Fig. 3 | Spectra from three independent 1D RCTE models and their residuals, fit to the weighted-average WASP-39b G395H transmission spectrum.** **a**, Spectra from the three models. **b**, Their residuals. The models are dominated by absorption from $H_2O$ and $CO_2$ with a grey-cloud-top pressure corresponding to ≈1 mbar. The models find that the data are best explained by 3–10 times solar metallicity (M/H) and sub-solar to solar C/O (C/O = 0.30–0.46). The extra absorption owing to $SO_2$, seen in the spectrum around 4.1 μm, is not included in the RCTE model grids and causes a marked impact on the $\chi^2/N$ (see Fig. 4).

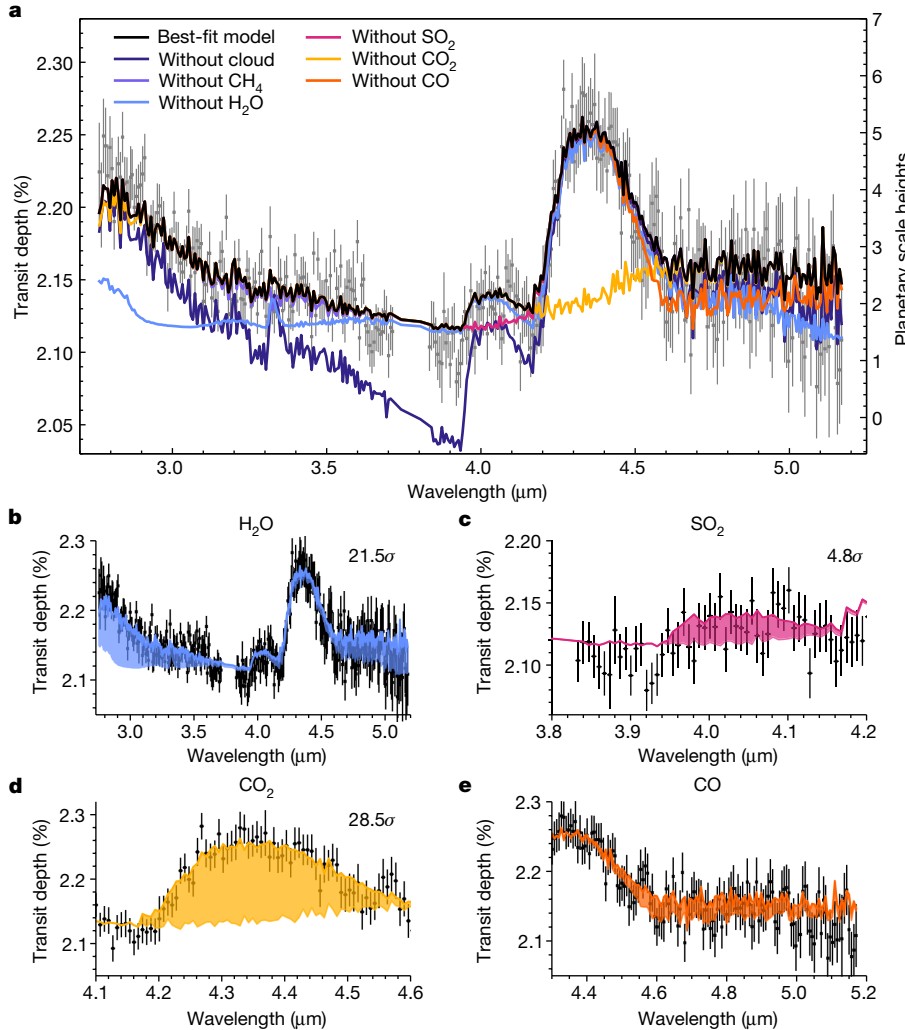

**Fig. 4 | Contribution of opacity sources to the best-fitting model with injected SO₂. a**, The lowest $\chi^2/N$ best-fitting model (PICASO in Fig. 3) with an injected abundance of $10^{-5.6}$ (VMR) $SO_2$. We also show this model with a selection of the anticipated absorbing species and the cloud opacity removed to indicate their contributions to the model. The inclusion of $SO_2$ in the model decreases the $\chi^2/N$ from 1.08 (shown in Fig. 3) to 1.02, resulting in a $4.8\sigma$ detection (see Extended Data Table 3). **b–e**, The effect of removing the corresponding molecular opacity from the spectrum (shaded region). Our best-fit model is also affected by minor opacities from CO, $H_2S$, OCS and $CH_4$, although their spectral features cannot be robustly detected in the spectrum. We show a model without CO and $CH_4$ in **a** to demonstrate this, with the minor contribution by CO also highlighted in **e**.

(ref. [36]) and/or photochemical destruction[34,37]. With regards to CO, $H_2S$ and OCS, we were unable to conclusively confirm their presence with these data. In particular, CO, $H_2O$, OCS and our modelled cloud deck all have overlapping opacity, which creates a pseudo-continuum from 4.6 to 5.1 µm (see Figs. 3 and 4). Therefore, we were unable to unambiguously identify the individual contributions from CO and other molecules over this wavelength region at the resolution presented in this work.

Our models show an atmosphere enriched in heavy elements, with best-fit parameters ranging from 3 to 10 times solar metallicity, given the spacing of individual model grids (see Methods). The spectra also indicate C/O ratios ranging from sub-solar to solar depending on the grid used, informed by the relative strength of absorption from carbon-bearing molecules to oxygen-bearing molecules. The interpretation of the relatively high resolution and precision of the G395H spectrum seems to be sensitive to the treatment of aerosols in the model, with one grid preferring 3 times solar metallicity when using a wavelength-dependent cloud opacity and physically motivated vertical cloud distribution[38] but 10 times solar metallicity when assuming a grey cloud. In general, forward model grids fit the main features of the data but do not place statistically significant constraints on many of the atmospheric parameters (see Methods). Future interpretation of the JTEC ERS WASP-39b data with Bayesian retrieval analyses will provide robust confidence intervals for these planetary properties and explore the degree to which these data are sensitive to modelling assumptions (for example, chemical equilibrium versus disequilibrium) and parameter degeneracies (for example, clouds versus atmospheric metallicity).

We are able to strongly rule out an absolute C/O ≥ 1 scenario ($\chi^2/N \geq 3.97$), which has previously been proposed for gas-dominated accretion at wide orbital radii beyond the $CO_2$ ice line at which the gas may be carbon-rich[39]. Our C/O upper limit, therefore, suggests that WASP-39b may have either formed at smaller orbital radii with gas-dominated accretion or that the accretion of solids enriched the atmosphere of WASP-39b with oxygen-bearing species[2]. The level of metal enrichment (3–10 times solar) is consistent with similar measurements of Jupiter and Saturn[40,41], potentially suggesting core-accretion formation scenarios[42], and is consistent with upper limits from interior-structure modelling[43]. These NIRSpec G395H transmission spectroscopy observations demonstrate the promise of robustly characterizing the atmospheric properties of exoplanets with JWST

unburdened by substantial instrumental systematics, unravelling the nature and origins of exoplanetary systems.

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

[1]School of Physics, HH Wills Physics Laboratory, University of Bristol, Bristol, UK. [2]Earth and Planets Laboratory, Carnegie Institution for Science, Washington, DC, USA. [3]NASA Ames Research Center, Moffett Field, CA, USA. [4]Department of Physics, Utah Valley University, Orem, UT, USA. [5]Center for Astrophysics | Harvard & Smithsonian, Cambridge, MA, USA. [6]Anton Pannekoek Institute for Astronomy, University of Amsterdam, Amsterdam, The Netherlands. [7]Department of Physics & Astronomy, University of Kansas, Lawrence, KS, USA. [8]Astrophysics Section, Jet Propulsion Laboratory, California Institute of Technology, Pasadena, CA, USA. [9]Department of Astrophysical Sciences, Princeton University, Princeton, NJ, USA. [10]Space Telescope Science Institute, Baltimore, MD, USA. [11]Department of Physics & Astronomy, Johns Hopkins University, Baltimore, MD, USA. [12]Department of Astronomy, Cornell University, Ithaca, NY, USA. [13]Carl Sagan Institute, Cornell University, Ithaca, NY, USA. [14]School of Earth and Planetary Sciences (SEPS), National Institute of Science Education and Research (NISER), Homi Bhabha National Institute (HBNI), Jatani, India. [15]Division of Geological and Planetary Sciences, California Institute of Technology, Pasadena, CA, USA. [16]Center for Space and Habitability, University of Bern, Bern, Switzerland. [17]Max Planck Institute for Astronomy, Heidelberg, Germany. [18]Department of Physics, Université de Montréal, Montreal, Quebec, Canada. [19]Institute for Research on Exoplanets, Université de Montréal, Montreal, Quebec, Canada. [20]Department of Astronomy and Astrophysics, University of California, Santa Cruz, Santa Cruz, CA, USA. [21]Department of Astronomy and Astrophysics, University of Chicago, Chicago, IL, USA. [22]Department of Astrophysical and Planetary Sciences, University of Colorado, Boulder, CO, USA. [23]European Space Agency, Space Telescope Science Institute, Baltimore, MD, USA. [24]Department of Physics and Astronomy, University College London, London, UK. [25]NASA Goddard Space Flight Center, Greenbelt, MD, USA. [26]Center for Computational Astrophysics, Flatiron Institute, New York, NY, USA. [27]School of Physics, Trinity College Dublin, Dublin, Ireland. [28]School of Earth and Space Exploration, Arizona State University, Tempe, AZ, USA. [29]University Observatory Munich, Ludwig Maximilian University of Munich, Munich, Germany. [30]Exzellenzcluster Origins, Garching, Germany. [31]Lunar and Planetary Laboratory, University of Arizona, Tucson, AZ, USA. [32]Instituto de Astrofísica de Canarias (IAC), Tenerife, Spain. [33]Departamento de Astrofísica, Universidad de La Laguna (ULL), Tenerife, Spain. [34]INAF - Palermo Astronomical Observatory, Palermo, Italy. [35]Space Science Institute, Boulder, CO, USA. [36]Steward Observatory, University of Arizona, Tucson, AZ, USA. [37]Department of Earth and Planetary Sciences, Johns Hopkins University, Baltimore, MD, USA. [38]Johns Hopkins University Applied Physics Laboratory, Laurel, MD, USA. [39]Atmospheric, Oceanic and Planetary Physics, Department of Physics, University of Oxford, Oxford, UK. [40]Indian Institute of Technology Indore, Indore, India. [41]Centre for Exoplanets and Habitability, University of Warwick, Coventry, UK. [42]Department of Physics, University of Warwick, Coventry, UK. [43]School of Physical Sciences, The Open University, Milton Keynes, UK.

[44]Bay Area Environmental Research Institute, NASA Ames Research Center, Moffett Field, CA, USA. [45]Department of Physics, New York University Abu Dhabi, Abu Dhabi, United Arab Emirates. [46]Center for Astro, Particle, and Planetary Physics (CAP3), New York University Abu Dhabi, Abu Dhabi, United Arab Emirates. [47]School of Physics and Astronomy, University of Leicester, Leicester, UK. [48]Centre for Exoplanet Science, University of St Andrews, St Andrews, UK. [49]Leiden Observatory, University of Leiden, Leiden, The Netherlands. [50]INAF - Osservatorio Astrofisico di Torino, Pino Torinese, Italy. [51]Space Research Institute, Austrian Academy of Sciences, Graz, Austria. [52]Institute of Astronomy, Department of Physics and Astronomy, KU Leuven, Leuven, Belgium. [53]Planetary Sciences Group, Department of Physics, University of Central Florida, Orlando, FL, USA. [54]Florida Space Institute, University of Central Florida, Orlando, FL, USA. [55]Universitäts-Sternwarte, Ludwig-Maximilians-Universität München, Munich, Germany. [56]Institute for Astrophysics, University of Vienna, Vienna, Austria. [57]Department of Astronomy, University of Maryland, College Park, MD, USA. [58]Department of Physics, Imperial College London, London, UK. [59]California Institute of Technology, Pasadena, CA, USA. [60]Laboratoire d'Astrophysique de Bordeaux, Université de Bordeaux, Pessac, France. [61]Département d'Astronomie, Université de Genève Sauverny, Versoix, Switzerland. [62]Department of Astronomy, University of Michigan, Ann Arbor, MI, USA. [63]Department of Physics, University of Rome "Tor Vergata", Rome, Italy. [64]INAF - Turin Astrophysical Observatory, Pino Torinese, Italy. [65]Department of Physics and Astronomy, Faculty of Environment, Science and Economy, University of Exeter, Exeter, UK. [66]SRON Netherlands Institute for Space Research, Leiden, The Netherlands. [67]Université Côte d'Azur, Observatoire de la Côte d'Azur, CNRS, Laboratoire Lagrange, Nice, France. [68]Department of Earth, Atmospheric and Planetary Sciences, Massachusetts Institute of Technology, Cambridge, MA, USA. [69]Kavli Institute for Astrophysics and Space Research, Massachusetts Institute of Technology, Cambridge, MA, USA. [70]Astronomy Department, Wesleyan University, Middletown, CT, USA. [71]Van Vleck Observatory, Wesleyan University, Middletown, CT, USA. [72]Institute of Astronomy, University of Cambridge, Cambridge, UK. [73]Maison de la Simulation, CEA, CNRS, Université Paris-Sud, Université Versailles St Quentin, Université Paris-Saclay, Gif-sur-Yvette, France. [74]Planetary Science Institute, Tucson, AZ, USA. [75]Université de Paris Cité and Université Paris-Est Creteil, CNRS, LISA, Paris, France. [76]Department of Earth and Planetary Sciences, University of California, Santa Cruz, Santa Cruz, CA, USA. [✉]e-mail: lili.alderson@bristol.ac.uk; hannah.wakeford@bristol.ac.uk

## Methods

### Data reduction

We produced several analyses of stellar spectra from the Stage 1 2D spectral images produced using the default STScI JWST Calibration Pipeline[44] ('rateints' files) and by means of customized runs of the STScI JWST Calibration Pipeline with user-defined inputs and processes for steps such as the 'jump detection' and 'bias subtraction' steps.

Each pipeline starts with the raw 'uncal' 2D images that contain group-level products. As we noticed that the default superbias images were of poor quality, we produced two customized runs of the JWST Calibration Pipeline, using either the default bias step or a customized version. The customized step built a pseudo-bias image by computing the median pixel value in the first group across all integrations and then subtracted the new bias image from all groups. We note that the poor quality of the default superbias images affects NRS1 more notably than NRS2, and this method could be revised once a better superbias is available.

Before ramp fitting, both our standard and custom bias step runs of the edited JWST Calibration Pipeline 'destriped' the group-level images to remove so-called '1/$f$ noise' (correlated noise arising from the electronics of the readout pattern, which appears as column striping in the subarray images[11,12]). Our group-level destriping step used a mask of the trace 15$\sigma$ from the dispersion axis for all groups within an integration, ensuring that a consistent set of pixels is masked within a ramp. The median values of non-masked pixels in each column were then computed and subtracted for each group.

The results of our customized runs of the JWST Calibration Pipeline are a set of custom group-level destriped products and custom bias-subtracted group-level destriped products. In both cases, the ramp-jump detection threshold of the JWST Calibration Pipeline was set to 15$\sigma$ (as opposed to the default of 4$\sigma$), as it produced the most consistent results at the integration level. In both custom runs of the JWST Calibration Pipeline, all other steps and inputs were left at the default values.

For all analyses, wavelength maps from the JWST Calibration Pipeline were used to produce wavelength solutions, verified against stellar absorption lines, for both detectors. The mid-integration times in $BJD_{TDB}$ were extracted from the image headers for use in producing light curves. None of our data-reduction pipelines performed a flat-field correction, as the available flat fields were of poor quality and unexpectedly removed portions of the spectral trace. In general, we found that 1/$f$ noise can be corrected at either the group or integration levels to similar effect; however, correction at the group level with a repeated column-by-column cleaning step at the integration level probably results in cleaner 1D stellar spectra. This was particularly true for NRS2, owing to the limited number of columns in which the unilluminated region on the detector extends both above and below the spectral trace, as shown in Extended Data Fig. 1.

Below we detail each of the independent data-reduction pipelines used to produce the time series of stellar spectra from our G395H observations.

**ExoTiC-JEDI pipeline.** We used the Exoplanet Timeseries Characterisation - JWST Extraction and Diagnostics Investigator (ExoTiC-JEDI[45]) pipeline on our custom group-level destriped products, treating each detector separately. Using the data-quality flags produced by the JWST Calibration Pipeline, we replaced any pixels identified as bad, saturated, dead, hot, low quantum efficiency or no gain value with the median value of surrounding pixels. We also searched each integration for pixels that were spatial outliers from the median of the surrounding 20 pixels in the same row by 6$\sigma$ (to remove permanently affected 'bad' pixels) or outliers from the median of that pixel in the surrounding ten integrations in time by 20$\sigma$ (to identify high-energy short-term effects such as cosmic rays) and replaced the outliers with the median

values. To obtain the trace position and FWHM, we fitted a Gaussian to each column of an integration, finding a median standard deviation of 0.7 pixels. A fourth-order polynomial was fitted through the trace centres and the widths, which were smoothed with a median filter, to obtain a simple aperture region. This region extended 5 times the FWHM of the spectral trace, above and below the centre, corresponding to a median aperture width of 7 pixels. To remove any remaining 1/$f$ and background noise from each integration, we subtracted the median of the unilluminated region in each column by masking all pixels that were 5 pixels away from the aperture. For each integration, the counts in each row and column of the aperture region were summed using an intrapixel extraction, taking the relevant fractional flux of the pixels at the edge of the aperture and cross-correlated to produce $x$-pixel and $y$-pixel shifts for detrending (see Extended Data Fig. 2). On average, the $x$-pixel shift represents movement of $1 \times 10^{-4}$ and $8 \times 10^{-6}$ of a pixel for NRS1 and NRS2, respectively. The aperture column sums resulted in 1D stellar spectra with errors calculated from photon noise after converting from data numbers using the gain factor. This reduction is denoted hereafter as ExoTiC-JEDI [V1].

We produced further 1D stellar spectra from both the custom group-level destriped product and custom bias-subtracted group-level destriped products using the ExoTiC-JEDI pipeline as described above, but with further cleaning by repeating the spatial outliers step. The reduction with further cleaning using the custom group-level destriped products is hence denoted as ExoTiC-JEDI [V2] and the reduction with further cleaning using the custom bias-subtracted group-level destriped products is hence denoted as ExoTiC-JEDI [V3].

**Tiberius pipeline.** We used the Tiberius pipeline, which builds on the LRG-BEASTS spectral reduction and analysis pipelines[15,46,47], on our custom group-level destriped products. For each detector, we created bad-pixel masks by manually identifying hot pixels in the data. We then combined them with pixels flagged as greater than 3$\sigma$ above the defined background. Before identifying the spectral trace, we interpolated each column of the detectors onto a grid 10 times finer than the initial spatial resolution. This step reduces the noise in the extracted data by improving the extraction of flux at the sub-pixel level, particularly where the edges of the photometric aperture bisect a pixel. We also interpolated over the bad pixels using their nearest-neighbouring pixels in $x$ and $y$.

We traced the spectra by fitting Gaussians at each column and used a running median, calculated with a moving box with a width of five data points, to smooth the measured centres of the trace. We fitted these smoothed centres with a fourth-order polynomial, removed points that deviated from the median by 3$\sigma$ and refitted with a fourth-order polynomial. To remove any residual background flux not captured by the group-level destriping, we fitted a linear polynomial along each column, masking the stellar spectrum. This was defined by an aperture with a width of 4 pixels centred on the trace. We also masked an extra 7 pixels on either side of the aperture so that the background was not fitting the wings of the stellar PSF and we clipped any pixels in the background that deviated by more than 3$\sigma$ from the mean for that particular column and frame. After removing the background in each column, the stellar spectra were then extracted by summing within a 4-pixel-wide aperture and correcting for pixel oversampling caused by the interpolation onto a finer grid, as described above. The uncertainties in the stellar spectra were calculated from the photon noise before background subtraction.

**transitspectroscopy pipeline.** We used the transitspectroscopy pipeline[48] on the 'rateints' products of the JWST Calibration Pipeline, treating each detector separately. The trace position was found from the median integration by cross-correlating each column with a Gaussian function, removing outliers using a median filter with a 10-pixel-wide window and smoothing the trace with a spline. We removed 1/$f$ noise from the 'rateints' products by masking all pixels within 10 pixels from

the centre of the trace and calculating and removing the median value from all columns. We then used optimal extraction[49] to obtain the 1D stellar spectra, with a 5-pixel-wide aperture above and below the trace. This allowed us to treat bad pixels and cosmic rays that had not been accounted for or masked in the 'rateints' products in an automated fashion. To monitor systematic trends in the observations, we also calculated the trace centre as described above and the FWHM for all integrations. The FWHM was calculated at each column and at each integration by first subtracting each column to half the maximum value in it, with a spline used to interpolate the profile. The roots of this profile were then found to estimate the FWHM.

**Eureka! pipeline.** We used two customized versions of the Eureka! pipeline[50], which combines standard steps from the JWST Calibration Pipeline with an optimal extraction scheme to obtain the time series of stellar spectra.

The first Eureka! reduction used our custom group-level destriped products and applied Stages 2 and 3 of Eureka! Stage 2, a wrapper of the JWST Calibration Pipeline, followed the default settings up to the flat fielding and photometric calibration steps, which were both skipped. Stage 3 of Eureka! was then used to perform the background subtraction and extraction of the 1D stellar spectra. We started by correcting for the curvature of NIRSpec G395H spectra by shifting the detector columns by whole pixels, to bring the peak of the distribution of the counts in each column to the centre of our subarray. To ensure that this curvature correction was smooth, we computed the shifts in each column for each integration from the median integration frame in each segment and applied a running median to the shifts obtained for each column. The pixel shifts were applied with periodic boundary conditions, such that pixels shifted upwards from the top of the subarray appeared at the bottom after the correction, ensuring no pixels were lost. We applied a column-by-column background subtraction by fitting and subtracting a flat line to each column of the curvature-corrected data frames, obtained by fitting all pixels further than six pixels from the central row. We also performed a double iteration of outlier rejection in time with a threshold of $10\sigma$, along with a $3\sigma$ spatial outlier-rejection routine, to ensure that bad pixels were not biasing our background correction. These outlier-rejection thresholds were selected to remove clear outliers in the data and provide a balance with the background subtraction step. We performed optimal extraction using an extraction profile defined from the median frame, the central nine rows of our subarray (four rows on either side of the central row). We also measured the vertical shift in pixels of the spectrum from one integration to the other using cross-correlation and the average PSF width at each integration, obtained by adding all columns together and fitting a Gaussian to the profile to estimate its width. This reduction is henceforth denoted as Eureka! [V1].

The second Eureka! reduction (Eureka! [V2]) used the 'rateints' outputs of the JWST Calibration Pipeline and applied Stage 2 of Eureka! as described above, with a modified version of Stage 3. In this reduction, we corrected the curvature of the trace using a spline and found the centre of the trace using the median of each column. We removed $1/f$ noise by subtracting the mean from each column, excluding the region 6 pixels away from the trace, sigma-clipping outliers at $3\sigma$. We extracted the 1D stellar spectra using a 4-pixel-wide aperture on either side of the trace centre.

## Limb-darkening
Limb-darkening is a function of the physical structure of the star that results in variations in the specific intensity of the light from the centre of the star to the limb, such that the limb looks darker than the centre. This is because of the change in depth of the stellar atmosphere being observed. At the limb of the star, the region of the atmosphere being observed at slant geometry is at higher altitudes and lower density, and thus lower temperatures, compared with the deeper atmosphere

observed at the centre of the star, at which hotter, denser layers are observed. The effect of limb-darkening is most prominent at shorter wavelengths, resulting in a more U-shaped light curve compared with the flat-bottomed light curves observed at longer wavelengths. To account for the effects of limb-darkening on the time-series light curves, we used analytical approximations for computing the ratio of the mean intensity to the central intensity of the star. The most commonly used limb-darkening laws for exoplanet transit light curves are the quadratic, square-root and nonlinear four-parameter laws[51]:

Quadratic:

$$\frac{I(\mu)}{I(1)} = 1 - u_1(1-\mu) - u_2(1-\mu)^2$$

Square-root:

$$\frac{I(\mu)}{I(1)} = 1 - s_1(1-\mu) - s_2(1-\sqrt{\mu})$$

Nonlinear four-parameter:

$$\frac{I(\mu)}{I(1)} = 1 - c_1(1-\mu^{0.5}) - c_2(1-\mu) - c_3(1-\mu^{1.5}) - c_4(1-\mu^2) \qquad (1)$$

in which $I(1)$ is the specific intensity in the centre of the disk, $u_1$, $u_2$, $s_1$, $s_2$, $c_1$, $c_2$, $c_3$ and $c_4$ are the limb-darkening coefficients and $\mu = \cos(\gamma)$, in which $\gamma$ is the angle between the line of sight and the emergent intensity.

The limb-darkening coefficients depend on the particular stellar atmosphere and therefore vary from star to star. For consistency across all of the light-curve fitting, we used 3D stellar models[52] for $T_{eff}$ = 5,512 K, $\log(g)$ = 4.47 cgs and Fe/H = 0.0, along with the instrument throughput to determine $I$ and $\mu$. As instrument commissioning showed that the throughput was higher than the pre-mission expectations[53], a custom throughput was produced from the median of the measured ExoTiC-JEDI [V2] stellar spectra, divided by the stellar model and Gaussian smoothed.

For the limb-darkening coefficients that were held fixed, we used the values computed using the ExoTiC-LD[54,55] package, which can compute the linear, quadratic and three-parameter and four-parameter nonlinear limb-darkening coefficients[51,56]. To compute and fit for the coefficients from the square-root law, we used previously outlined formalisms[57,58]. We highlight that we do not see any dependence in our transmission spectra on the limb-darkening procedure used across our independent reductions and analyses.

## Light-curve fitting
From the time series of extracted 1D stellar spectra, we created our broadband transit light curves by summing the flux over 2.725–3.716 μm for NRS1 and 3.829–5.172 μm for NRS2. For the spectroscopic light curves, we used a common 10-pixel binning scheme within these wavelength ranges to generate a total of 349 spectroscopic bins (146 for NRS1 and 203 for NRS2). We also tested wider and narrower binning schemes but found that 10-pixel-wide bins achieved the best compromise between the noise in the spectrum and showcasing the abilities of G395H across analyses. In our analyses, we treated the NRS1 and NRS2 light curves independently to account for differing systematics across the two detectors. To construct the full NIRSpec G395H transmission spectrum of WASP-39b, we fitted the NRS1 and NRS2 broadband and spectroscopic light curves using 11 independent light-curve-fitting codes, which are detailed below. When starting values were required, all analyses used the same system parameters[37]. In many of our analyses, we detrended the raw broadband and spectroscopic light curves using the time-dependent decorrelation parameters for the change in the FWHM of the spectral trace or the shift in x-pixel and y-pixel positions (Extended Data Fig. 2). We also used various approaches to account for

the mirror-tilt event, which we found to have a smaller effect at longer wavelengths (Extended Data Fig. 3).

Using fitting pipeline 1, we measured a centre of transit time ($T_0$) of $T_0 = 2,459,791.612039 \pm 0.000017$ BJD$_{TDB}$ and $T_0 = 2,459,791.6120689 \pm 0.000021$ BJD$_{TDB}$ computed from the NRS1 and NRS2 broadband light curves, respectively; most of the fitting pipelines obtained $T_0$ within $1\sigma$ of the quoted uncertainty.

For each of our analyses, we computed the expected photon noise from the raw counts taking into account the instrument read noise (16.18 e$^-$ on NRS1 and 17.75 e$^-$ on NRS2), gain (1.42 for NRS1 and 1.62 for NRS2) and the background counts (which are found to be negligible after cleaning) and compared it to the final signal-to-noise ratio in our light curves (see Fig. 1). We also determine the level of white and red noise in our spectroscopic light curves by computing the Allan deviation[59], which is used to measure the deviation from the expected photon noise by binning the data into successively smaller bins (that is, fewer data points per bin) and calculating the signal-to-noise ratio achieved[60]. Extended Data Fig. 4 shows the Allan deviation for three of the 11 reductions performed on the data (see the ExoTiC-ISM noise_calculator function[54]).

Although there is a general consensus across each of the data analyses, by comparing the results of each fitting pipeline, we were better able to evaluate the impact of different approaches to the data reduction, such as the removal of bad pixels. For future studies, we recommend the application of several pipelines that use differing analysis methods, such as the treatment of limb-darkening, systematic effects and noise removal. No single pipeline presented on its own can fully evaluate the measured impact of each effect, given the differing strategies, targets and potential for chance events such as a mirror tilt with each observation. In particular, attention should be paid to $1/f$ noise removal at the group versus integration levels for observations with fewer groups per integration than this study.

Below, we detail each of our 11 fitting pipelines and summarise them in Extended Data Table 1.

**Fitting pipeline 1: ExoTiC-JEDI.** We fitted the broadband and spectroscopic light curves produced from the ExoTIC-JEDI [V3] stellar spectra using the least-squares optimizer, scipy.optimize lm (ref. [61]). We simultaneously fitted a batman transit model[62] with a constant baseline and systematics models for data pre-tilt and post-tilt event, fixing the centre of transit time $T_0$, the ratio of the semi-major axis to stellar radius $a/R_\star$ and the inclination $i$ to the broadband light-curve best-fit values. The systematics models included a linear regression on $x$ and $y$, for which $x$ and $y$ are the measured trace positions in the dispersion and cross-dispersion directions, respectively. We accounted for the tilt event by normalizing the light curve pre-tilt by the median pre-transit flux and normalizing the light curve post-tilt by the median post-transit flux. We discarded the first 15 integrations and the three integrations during the tilt event. Fourteen-pixel columns were discarded owing to outlier pixels directly on the trace. We fixed the limb-darkening coefficients to the four-parameter nonlinear law.

**Fitting pipeline 2: Tiberius.** We used the broadband light curves generated from the Tiberius stellar spectra and fitted for the ratio of the planet to stellar radii $R_p/R_\star$, as well as $i$, $T_0$, $a/R_\star$, the quadratic law limb-darkening coefficient $u_1$ and the systematics model parameters, the $x$-pixel and $y$-pixel shifts, FWHM and sky background, with the period $P$, the eccentricity $e$ and $u_2$ fixed. We used uniform priors for all the fitted parameters. Our analytic transit light-curve model was generated with batman. We fitted our broadband light curve with a transit + systematics model using a Gaussian process (GP)[63,64], implemented through george[65], and a Markov chain Monte Carlo method, implemented through emcee[66]. For our Tiberius spectroscopic light curves, we held $a/R_\star$, $i$ and $T_0$ fixed to the values determined from the broadband light-curve fits and applied a systematics correction from the broadband light-curve fit to aid in fitting the mirror-tilt event. We

fitted the spectroscopic light curves using a GP with an exponential squared kernel for the same systematics detrending parameters detailed above. We used a Gaussian prior for $a/R_\star$ and uniform priors for all other fitted parameters.

**Fitting pipeline 3: Aesop.** We used transit light curves from the ExoTiC-JEDI [V1] stellar spectra and fit the broadband and spectroscopic light curves using the least-squares minimizer LMFIT[67]. We fitted each light curve with a two-component function consisting of a transit model (generated using batman) multiplied by a systematics model. Our systematics model included the $x$-pixel and $y$-pixel positions on the detector ($x$, $y$, $xy$, $x^2$ and $y^2$). To capture the amplitude of the tilt event in our systematics model, we carried out piecewise linear regression on the out-of-transit baseline pre-tilt and post-tilt. We first fit the broadband light curve by fixing $P$ and $e$ and fitting for $T_0$, $a/R_\star$, $i$, $R_p/R_\star$, stellar baseline flux and systematic trends using wide uniform priors. For the spectroscopic light curves, we fixed $T_0$, $a/R_\star$ and $i$ to the best-fit values from the broadband light curve and fit for $R_p/R_\star$. We held the nonlinear limb-darkening coefficients fixed.

**Fitting pipeline 4: transitspectroscopy.** We fit the broadband and spectroscopic light curves produced from the transitspectroscopy stellar spectra, running juliet[68] in parallel with the light-curve-fitting module of the transitspectroscopy pipeline[48] with dynamic nested sampling through dynesty[69] and analytical transit models computed using batman. We fit the broadband light curves for NRS1 and NRS2 individually, fixing $P$, $e$ and $\omega$ and fitting for the impact parameter $b$, as well as $T_0$, $a/R_\star$, $R_p/R_\star$, extra jitter and the mean out-of-transit flux. We also fitted two linear regressors, a simple slope and a 'jump' (a regressor with zeros before the tilt event and ones after the tilt event), scaled to fit the data. We fitted the square-root-law limb-darkening coefficients using the Kipping sampling scheme. We fitted the spectroscopic light curves at the native resolution of the instrument, fixing $T_0$, $a/R_\star$ and $b$. We used the broadband light-curve systematics model for the spectroscopic light curve, with wide uniform priors for each wavelength bin, and set truncated normal priors for the square-root-law limb-darkening coefficients. We also fitted a jitter term added in quadrature to the error bars at each wavelength with a log-uniform prior between 10 and 1,000 ppm. We computed the mean of the limb-darkening coefficients by first computing the nonlinear coefficients from ATLAS models[70] and passing them through the SPAM algorithm[71]. We binned the data into 10-pixel-wavelength bins after fitting the native-resolution light curves.

**Fitting pipeline 5: ExoTEP.** We fitted the transit light curves from the Eureka! [V1] stellar spectra using the ExoTEP analysis framework[72–75]. ExoTEP uses batman to generate analytical light-curve models, adds an analytical instrument systematics model along with a photometric scatter parameter and fits for the best-fit parameters and their uncertainties using emcee. Before fitting, we cleaned the light curves by running ten iterations of $5\sigma$ clipping using a running median of window length 20 on the flux, $x$-pixel and $y$-pixel shifts and the 'ydriftwidth' data product from Eureka! Stage 3 (the average spatial PSF width at each integration). Our systematics model consisted of a linear trend in time with a 'jump' (constant offset) after the tilt event. The 'ydriftwidth' was used before the fit to locate the tilt event. We used a running median of 'ydriftwidth' to search for the largest offset and flagged every data point after the tilt event so that they would receive a constant 'jump' offset in our systematics model. We also removed the first point of the tilt event in our fits, as it was not captured by the 'jump' model. We fitted the broadband light curves, fitting for $R_p/R_\star$, photometric scatter, $T_0$, $b$, $a/R_\star$, the quadratic limb-darkening coefficients and the systematics model parameters (normalization constant, slope in time and constant 'jump' offset). We used uninformative flat priors on all the parameters. The orbital parameters were fixed to the best-fit broadband light curve values for the subsequent spectroscopic light-curve fits.

**Fitting pipeline 6.** We fitted transit light curves from the ExoTiC-JEDI [V1] stellar spectra using a custom lmfit light-curve-fitting code. The final systematic detrending model included a batman analytical transit model multiplied by a systematics model consisting of a linear stellar baseline term, a linear term for the $x$-pixel and $y$-pixel shifts and an exponential ramp function. The tilt event was accounted for by decorrelating the light curves with the $y$-pixel shifts, using a (1 + constant × $y$-shift) term with the constant fitted for in each light curve. For the broadband light-curve fits, we fixed $P$ and fitted for $T_0$, $i$, $R_p/R_\star$, $a/R_\star$, $x$-pixel and $y$-pixel shifts and the exponential ramp amplitude and timescale. We fixed the nonlinear limb-darkening coefficients. For the spectroscopic light-curve fits, we fixed the values of $T_0$, $i$ and $a/R_\star$ and the exponential ramp timescale to the broadband light-curve-fit values, and fitted for $R_p/R_\star$, the $x$-pixel and $y$-pixel shifts and the ramp amplitude. Wide, uniform priors were used on all the fitting parameters in both the broadband and spectroscopic light-curve fits.

**Fitting pipeline 7.** We fitted transit light curves from the Eureka! [V2] stellar spectra, using PyLightcurve (ref. [75]) to generate the transit model with emcee as the sampler. We calculated the nonlinear four-parameter limb-darkening coefficients using ExoTHETyS (ref. [76]), which relies on PHOENIX 2012–2013 stellar models[77,78], and fixed these in our fits to the precomputed theoretical values. Our full transit + systematics model included a transit model multiplied by a second-order polynomial in the time domain. We accounted for the tilt event by subtracting the mean of the last 30 integrations of the pre-transit data from the mean of the first 30 integrations of the post-transit data, to account for the jump in flux, shifting the post-transit light curve upwards by the jump value. We fitted for the systematics (the parameters of the second-order polynomial), $R_p/R_\star$ and $T_0$. We used uniform priors for all the fitted parameters. We adopted the root mean square of the out-of-transit data as the error bars for the light-curve data points to account for the scatter in the data.

**Fitting pipeline 8.** We used the transit light curves generated from the ExoTiC-JEDI [V1] stellar spectra. We fit the broadband light curves with a batman transit model multiplied by a second-order systematics model as a function of $x$-pixel and $y$-pixel shifts. We fixed both of the quadratic limb-darkening coefficients for each wavelength bin. We fitted for $R_p/R_\star$, $i$, $T_0$ and $a/R_\star$, using wide uninformed priors, and ran our fits using emcee. For the spectroscopic light-curve fits, we fixed $i$ and $a/R_\star$ to the broadband light-curve best-fit values and fitted for an extra error term added in quadrature.

**Fitting pipeline 9.** We used the transit light curves from the ExoTiC-JEDI [V1] stellar spectra. We fixed both of the quadratic limb-darkening coefficients and fitted the light curves with a batman transit model multiplied by a systematics model of a second-order function of $x$-pixel and $y$-pixel shifts. We fixed the best-fit broadband light-curve values for $T_0$, $a/R_\star$ and $i$ for the spectroscopic light-curve fits and fitted for $R_p/R_\star$ using emcee for each 10-pixel bin, with the walkers initialized in a tight cluster around the best-fit solution from a Levenberg–Marquardt minimization. For both the broadband and spectroscopic light curves, we also fit for an extra per-point error term.

**Fitting pipeline 10.** We fitted the transit light curves from the ExoTiC-JEDI [V2] stellar spectra and performed our model fitting using automatic differentiation implemented with JAX (ref. [79]). We used a GP systematics model with a time-dependent Matérn ($\nu = 3/2$) kernel and a variable white-noise jitter term. The mean function consists of a linear trend in time plus a sigmoid function to account for the drop in measured flux that occurred mid-transit owing to the mirror-tilt event. For the transit model, we used the exoplanet package[80], making use of previously developed light-curve models[81,82].

For the GP systematics component, a generalization of the algorithm used by the celerite package[83] was adapted for JAX. We fixed both of the quadratic limb-darkening coefficients. For the initial broadband light-curve fit, both NRS1 and NRS2 were fitted simultaneously. All transit parameters were shared across both light curves, except for $R_p/R_\star$, which was allowed to vary for NRS1 and NRS2 independently. We fitted for $T_0$, the transit duration $b$ and both $R_p/R_\star$ values. For the spectroscopic light-curve fits, all transit parameters were then fixed to the maximum-likelihood values determined from the broadband fit, except for $R_p/R_\star$, which was allowed to vary for each wavelength bin. Uncertainties for the transit model parameters, including $R_p/R_\star$, were assumed to be Gaussian and estimated using the Fisher information matrix at the location of the maximum-likelihood solution, which was computed exactly using the JAX automatic differentiation framework.

**Fitting pipeline 11: Eureka!.** We used transit light curves from the Eureka! [V2] time-series stellar spectra with the open-source Eureka! code to estimate the best-fit transit parameters and their uncertainties using a Markov chain Monte Carlo method fit to the data (implemented by emcee). A linear trend in time was used as a systematics model and a step function was used to account for the tilt event. We fixed $a/R_\star$, $i$, $T_0$ and the time of the tilt event to the best-fit values from the NRS1 broadband light curve, with the three integrations during the tilt event clipped from the light curve. We fitted for $R_p/R_\star$, both quadratic limb-darkening coefficients, the linear time trend and the magnitude of the step from the tilt event, with uniform priors for both the magnitude of the step and the limb-darkening coefficients.

**Transmission spectral analysis**
On the basis of the independent light-curve fits described above, we produced 11 transmission spectra from our NIRSpec G395H observations using several analyses and fitting methods. Extended Data Table 1 shows a breakdown of the different steps used in each fitting pipeline. In this work, three different 2D spectral image products were used, producing seven different 1D stellar spectra. Eleven different fitting pipelines using five different limb-darkening methods were then applied. Each of these fitting pipelines resulted in an independent analysis of the observations and 11 comparative transmission spectra. Extended Data Fig. 5 details comparative information for all 11 analyses to quantify their similarities and differences.

We computed the standard deviation of the 11 spectra in each wavelength bin and compared this to the mean uncertainty obtained in that bin. The average standard deviation in each bin across all fitting pipelines was 199 ppm, compared with an average uncertainty of 221 ppm (which ranged from 131 to 625 ppm across the bins). The computed standard deviation in each bin across all pipelines ranged from 85 to 1,040 ppm, with greater than 98% of the bins having a standard deviation lower than 500 ppm. We see an increase in scatter at longer wavelengths, with the structure of the scatter following closely with the measured stellar flux, for which throughput beyond 3.8 μm combines with decreasing stellar flux. The unweighted mean uncertainty of all 11 transmission spectra follows a similar structure to the standard deviation, with the uncertainty increasing at longer wavelengths. The uncertainties from each fitting pipeline are consistent to within $3\sigma$ of each other, with the uncertainty per bin typically overestimated compared with the mean uncertainty across all reductions.

From all 11 transmission spectra, we computed a weighted-average transmission spectrum using the transit depth values from all reductions in each bin weighted by 1/variance ($1/\sigma^2$, in which $\sigma$ is the uncertainty on the data point from each reduction). For this weighted-average transmission spectrum, the unweighted mean of the uncertainties in each bin was used to represent the error bar on each point. By using the weighted average of all 11 independently obtained transmission spectra, we therefore do not apply infinite weight to any one reduction in our interpretation of the atmosphere. Although the weighted

average could be sensitive to any one spectrum with underestimated uncertainties, we find that our uncertainties are typically overestimated compared with the average. Similarly, we chose to use the mean rather than the median of the transmission spectral uncertainties, as this results in a more conservative estimate of the uncertainties in each bin. We find that all of the 11 transmission spectra are within $2.95\sigma$ of the weighted-average transmission spectrum without applying offsets.

We calculated normalized transmission spectrum residuals for each fitting pipeline by subtracting the weighted-average spectrum and dividing by the uncertainty in each bin. We generated histograms of the normalized transmission spectrum residuals and used the mean and standard deviation of the residuals to compute a normalized probability density function (PDF). We performed a Kolmogorov–Smirnov test on each of the normalized residuals and found that they are all approximately symmetric around their means, with normal distributions. This confirms that they are Gaussian in shape, with the null hypothesis that they are not Gaussian strongly rejected in the majority of cases (see Extended Data Fig. 5).

The PDFs of the residuals indicate three distinct clusters of computed spectra based on their deviations from the mean and their spreads. The first cluster is negatively offset by less than 200 ppm and corresponds to fitting pipelines that used extracted stellar spectra and that underwent further cleaning steps (for example, ExoTiC-JEDI [V3]). The second cluster is positively offset from the mean by about 120 ppm and contains most of the transmission spectra produced. We see no obvious trends in this group to any specific reduction or fitting process. The final cluster is centred around the mean but has a broad distribution, suggesting a larger scatter both above and below the average transmission spectrum. This is probably the result of uncleaned outliers or marginal offsets between NRS1 and NRS2. These transmission spectra demonstrate that the 11 independent fitting pipelines are able to accurately reproduce the same transmission spectral feature structures, further highlighting the minimized impact of systematics on the time-series light curves. We suspect that the minor differences resulting from different reduction products and fitting pipelines are linked to the superbias and treatment of $1/f$ noise. We anticipate that the impacts of these will be improved with new superbias images, expected to be released soon by STScI, and with more detailed investigation into the impact of $1/f$ noise at the group level beyond the scope of this work.

## Model comparison

To identify spectral absorption features, we compared the resulting weighted-average transmission spectrum of WASP-39b to several 1D RCTE atmosphere models from three independent model grids. Each forward model is computed on a set of common physical parameters (for example, metallicity, C/O ratio, internal temperature and heat redistribution), shown in Extended Data Table 2. Additionally, each model grid contains different prescriptions for treating certain physical effects (for example, scattering aerosols). Although each grid contains different opacity sources from varying line lists (see Extended Data Table 2), they each consider all of the main molecular and atomic species[84]. Each model transmission spectrum from the grids was binned to the same resolution as that of the observations to compute the $\chi^2$ per data point, with a wavelength-independent transit depth offset as the free parameter. In general, the forward model grids fit the main features of the data but are unable to place statistically significant constraints on many of the atmospheric parameters, owing to both the finite nature of the forward model grid spacing[13] and the insensitivity of some of these parameters to the 3–5-µm transmission spectrum of WASP-39b (for example, >100 K differences in interior temperature provided nearly identical $\chi^2/N$).

**ATMO.** We used the ATMO RCTE grid[85–88], which consists of model transmission spectra for different day–night energy redistribution factors, atmospheric metallicities, C/O ratios, haze factors and grey cloud factors with a range of line lists and pressure-broadening sources[88]. In total, there were 5,160 models. Within this grid, we find the best-fit model to have 3 times solar metallicity, with a C/O ratio of 0.35 and a grey cloud opacity 5 times the strength of $H_2$ Rayleigh scattering at 350 nm and a $\chi^2/N = 1.098$ for $N = 344$ data points and only fitting for an absolute altitude change in $y$.

**PHOENIX.** We calculated a grid of transmission spectra using the PHOENIX atmosphere model[89–91], varying the heat redistribution of the planet, atmospheric metallicity, C/O ratio, internal temperature, the presence of aerosols and the atmospheric chemistry (equilibrium or rainout). Opacities used include the BT2 $H_2O$ line list[92], as well as HITRAN for 129 other main molecular absorbers[93] and Kurucz and Bell data for atomic species[94]. The HITRAN line lists available in this version of PHOENIX are often complete only at room temperature, which may be the cause of the apparent shift in the $CO_2$ spectral feature compared with the other grids that primarily use HITEMP and ExoMol lists. This shift is the cause of the difference in $\chi^2$ between PHOENIX and the other model grids. In total, there were 1,116 models. Within this grid, the best-fit model has 10 times solar metallicity, a C/O ratio of 0.3, an internal temperature of 400 K, rainout chemistry and a cloud deck top at 0.3 mbar. The best-fit model has a $\chi^2/N = 1.203$ for $N = 344$ data points.

**PICASO 3.0 and Virga.** We used the open-source radiative–convective equilibrium model PICASO 3.0 (refs. [95,96]), which has its heritage in the Fortran-based EGP mode[97,98], to compute a grid of 1D pressure–temperature models for WASP-39b. The opacity sources included in PICASO 3.0 are listed in Extended Data Table 2. Of the 29 molecular opacity sources included, the line lists of notable molecules used were: $H_2O$ (ref. [99]), $CO_2$ (ref. [100]), $CH_4$ (ref. [101]) and CO (ref. [102]). The parameters varied in this grid of models include the interior temperature of the planet ($T_{int}$), atmospheric metallicity, C/O ratio and the dayside-to-nightside heat redistribution factor (see Extended Data Table 2), with correlated-$k$ opacities[98,103]. In total, there were 192 cloud-free models. We include the effect of clouds in two ways. First, we post-processed the pressure–temperature profile using the cloud model Virga[95,104], which follows from previously developed methodologies[38], in which we included three condensable species (MnS, $Na_2S$ and $MgSiO_3$). Virga requires a vertical mixing parameter, $K_{zz}$ (cm$^2$ s$^{-1}$), and a vertically constant sedimentation efficiency parameter, $f_{sed}$. In general, $f_{sed}$ controls the vertical extent of the cloud opacity, with low values ($f_{sed} < 1$) creating large, vertically extended cloud decks with small particle sizes. In total, there were 3,840 cloudy models. The best fit from our grid with Virga-computed clouds has 3 times solar metallicity, solar C/O (0.458) and $f_{sed} = 0.6$, which results in a $\chi^2/N = 1.084$.

As well as the grid fit, we also use the PICASO framework to quantify the feature-detection significance. In this method, we are able to incorporate clouds on the fly using the fitting routine PyMultiNest[105]. We fit for each of the grid parameters using a nearest-neighbour technique and a radius scaling to account for the unknown reference pressure, giving five parameters in total. When fitting for clouds, we either fit for $K_{zz}$ and $f_{sed}$ in the Virga framework (seven parameters in total) or we fit for the cloud-top pressure corresponding to a grey cloud deck with infinite opacity (six parameters in total). These results are described in the following section.

## Feature-detection significance

From the chemical equilibrium results of the single best-fit models, the molecules that could potentially contribute to the spectrum based on their abundances and 3–5-µm opacity sources are $H_2$ and He (via continuum) and CO, $H_2O$, $H_2S$, $CO_2$ and $CH_4$. More minor sources of opacity with VMR abundances <1 ppm are molecules such as OCS and $NH_3$. For example, removing $H_2S$, $NH_3$ and OCS from the single best-fit PICASO 3.0 model increases the chi-square value by less than 0.002.

Therefore, we focus on computing the statistical significance of only $H_2O$, $SO_2$, $CO_2$, $CH_4$ and CO.

To quantify the statistical significance, we performed two different tests. First, we used a Gaussian residual fitting analysis, as used in other JTEC ERS analyses[23,29,31]. In this method, we subtracted the best-fit model without a specific opacity source from the weighted-average spectrum of WASP-39b, isolating the supposed spectral feature. We then fit a three-parameter or four-parameter Gaussian curve to the residual data using a nested sampling algorithm to calculate the Bayesian evidence[106]. For $H_2O$ and CO, the extra transit depth offset parameter for the Gaussian fit was necessary to account for local mismatch of the fit to the continuum, whereas only a mean, standard deviation and scale parameter were required for a residual fit to the other molecules. We then compared this to the Bayesian evidence of a flat line to find the Bayes factor between a model that fits the spectral feature versus a model that excludes the spectral feature. These fits are shown in Extended Data Fig. 6.

Although the Gaussian residual fitting method is useful for quantifying the presence of potentially unknown spectral features, it cannot robustly determine the source of any given opacity. We therefore used the Bayesian fitting routine from PyMultiNest in the PICASO 3.0 framework to refit the grid parameters, while excluding the opacity contribution from the species in question. Then, we compared the significance of the molecule through a Bayes factor analysis[107]. Those values are shown in Extended Data Table 3.

We find significant evidence (>$3\sigma$) for $H_2O$, $CO_2$ and $SO_2$. In general, the two methods only agree well for molecules whose contribution has a Gaussian shape (that is, $SO_2$ and $CO_2$). For example, for $CO_2$, we find decisive $28.5\sigma$ and $26.9\sigma$ detections for the Bayes factor and Gaussian analysis, respectively. Similarly, for $H_2O$, we find $21.5\sigma$ and $16.5\sigma$ detections, respectively. The evidence for $SO_2$ is less substantial, but both methods give significant detections of $4.8\sigma$ and $3.5\sigma$, respectively. Although the Gaussian fitting method found a broad 1-µm-wide residual in the region of CO (that is, >4.5 µm), its shape was unlike that seen with the PRISM data[31]. CO remained undetected with the Bayesian fitting analysis and therefore we are unable to robustly confirm evidence of CO. Similarly, no evidence for $CH_4$ was found[23]. Gaussian residual fitting in the region of $CH_4$ absorption only found a very broad inverse Gaussian and so is not included in Extended Data Table 3.

## $SO_2$ absorption

We performed an injection test with the PICASO best-fit model in the PyMultiNest fitting framework to determine the abundance of $SO_2$ required to match the observations. We add $SO_2$ opacity using the ExoMol line list[108], without rerunning the RCTE model to self-consistently compute a new climate profile. Fitting for the cloud deck dynamically, without $SO_2$, produces a single best estimate of 10 times solar metallicity, sub-solar C/O (0.229), resulting in a marginally worse $\chi^2/N = 1.11$. With $SO_2$, the single best fit tends back to 3 times solar metallicity, solar C/O. This suggests that cloud treatment and the exclusion of spectrally active molecules have an effect on the resultant physical interpretation of bulk atmospheric parameters. Ultimately, if we fit for $SO_2$ in our PyMultiNest framework with the Virga cloud treatment, we obtain 3 times solar metallicity, solar C/O, log $SO_2 = -5.6 \pm 0.1$ ($SO_2 = 2.5 \pm 0.65$ ppm) and $\chi^2/N = 1.02$, which is our single best-fit model (shown in Fig. 4). For context, an atmospheric metallicity of 3–10 times solar would provide a thermochemical equilibrium abundance of 72–240 ppm $H_2S$, the presumed source for photochemically produced $SO_2$ (ref. [36]).

To confirm the plausibility of $SO_2$ absorption to explain the 4.1-µm spectral feature, we also computed models with prescribed, vertically uniform $SO_2$ VMRs of 0, 1, 5 and 10 ppm using the structure from the best-fit PHOENIX model (10 times solar metallicity, C/O = 0.3). We calculated ad hoc spectra using the gCMCRT radiative transfer code[109] with the ExoMol $SO_2$ line list[108] (see Extended Data Fig. 7). Linearly interpolating the models with respect to the $SO_2$ abundance

and performing a Levenberg–Marquardt regression gave a best-fit value of $4.6 \pm 0.67$ ppm. Inserting this abundance of $SO_2$ into the best-fit PHOENIX model improves the $\chi^2/N$ from 1.2 to 1.08.

Future atmospheric retrievals can provide a more statistically robust measurement for the $SO_2$ abundance and add extra information from the similar absorption seen in the PRISM transmission spectrum[29,31].

## 4.56-µm feature

A 0.08-µm-wide bump in transit depth centred at 4.56 µm is not fit by any of the model grids. This feature, picked up by the resolution of G395H, is not clearly seen in other ERS observations of WASP-39b. Following the same Gaussian residual fitting procedure as described above, we found a feature significance of $3.3\sigma$ (see Extended Data Fig. 6). To identify possible opacity sources in the atmosphere of WASP-39b that might be the cause of this absorption, we compared the feature with $CH_4$ (ref. [110]), $C_2H_2$ (ref. [111]), $C_2H_4$ (ref. [112]), $C_2H_6$ (ref. [113]), CO (ref. [114]), $CO_2$ (ref. [100]), $CS_2$ (ref. [113]), CN (ref. [115]), HCN (ref. [116]), HCl (ref. [113]), $H_2S$ (ref. [117]), HF (ref. [118]), $H_3^+$ (ref. [119]), LiCl (ref. [115]), $NH_3$ (ref. [120]), NO (ref. [114]), $NO_2$ (ref. [113]), $N_2O$ (ref. [114]), $N_2$ (ref. [121]), NaCl (ref. [122]), OCS (ref. [113]), $PH_3$ (ref. [123]), PN (ref. [124]), PO (ref. [125]), SH (ref. [126]), SiS (ref. [127]), $SiH_4$ (ref. [128]), SiO (ref. [129]), the X–X state of SO (ref. [130]), $SO_2$ (ref. [108]), $SO_3$ (ref. [108]) and isotopologues of $H_2O$, $CH_4$, $CO_2$ and CO, but did not find a convincing candidate that showed opacity at the correct wavelength or the correct width. The narrowness of the feature suggests that it could be a very distinct Q-branch, in which the rotational quantum number in the ground state is the same as the rotational quantum number in the excited state. However, of the molecules we explored, there were no candidates with a distinct Q-branch at this wavelength whose P-branch and R-branch did not obstruct the neighbouring $CO_2$ and continuum-like $CO + H_2O$ opacity.

We also note that many of these species lack high-temperature line-list data, making it difficult to definitively rule out such species. For example, OCS, SO and $CS_2$ are available in HITRAN2020 (ref. [113]) but not in ExoMol[131]. Furthermore, if photochemistry is important for WASP-39b, as indicated by the presence of $SO_2$, then there may be many species out of equilibrium that may contribute to the transit spectrum, some of which do not have high-temperature opacity data at present (such as OCS, $NH_2$ or HSO). Future observations over this wavelength region of this and other planets may confirm or refute the presence of this unknown absorber.

## Data availability

The data used in this paper are associated with JWST programme ERS 1366 (observation #4) and are available from the Mikulski Archive for Space Telescopes (MAST; https://mast.stsci.edu). Science data processing version (SDP_VER) 2022_2a generated the uncalibrated data that we downloaded from MAST. We used JWST Calibration Pipeline software version (CAL_VER) 1.5.3 with modifications described in the text. We used calibration reference data from context (CRDS_CTX) 0916, except as noted in the text. All the data and models presented in this publication can be found at https://doi.org/10.5281/zenodo.7185300.

## Code availability

The codes used in this publication to extract, reduce and analyse the data are as follows; STScI JWST Calibration Pipeline[44] (https://github.com/spacetelescope/jwst), Eureka![50] (https://eurekadocs.readthedocs.io/en/latest/), ExoTiC-JEDI[45] (https://github.com/Exo-TiC/ExoTiC-JEDI), juliet[68] (https://juliet.readthedocs.io/en/latest/), Tiberius[15,46,47], transitspectroscopy[48] (https://github.com/nespinoza/transitspectroscopy). Furthermore, these made use of batman[62] (http://lkreidberg.github.io/batman/docs/html/index.html), celerite[83] (https://celerite.readthedocs.io/en/stable/), chromatic (https://zkbt.github.io/chromatic/), dynesty[69] (https://dynesty.readthedocs.io/en/stable/

index.html), emcee[66] (https://emcee.readthedocs.io/en/stable/), exoplanet[80] (https://docs.exoplanet.codes/en/latest/), ExoTEP[72–74], ExoTHETyS[76] (https://github.com/ucl-exoplanets/ExoTETHyS), ExoTiC-ISM[54] (https://github.com/Exo-TiC/ExoTiC-ISM), ExoTiC-LD[55] (https://exotic-ld.readthedocs.io/en/latest/), george[65] (https://george.readthedocs.io/en/latest/), JAX[79] (https://jax.readthedocs.io/en/latest/), LMFIT[67] (https://lmfit.github.io/lmfit-py/), PyLightcurve[75] (https://github.com/ucl-exoplanets/pylightcurve), PyMC3 (ref. [132]) (https://docs.pymc.io/en/v3/index.html) and Starry[81] (https://starry.readthedocs.io/en/latest/), each of which use the standard Python libraries astropy[133,134], matplotlib[135], numpy[136], pandas[137], scipy[61] and xarray[138]. The atmospheric models used to fit the data can be found at ATMO[85–88], PHOENIX[89–91], PICASO[95,96] (https://natashabatalha.github.io/picaso/), Virga[95,104] (https://natashabatalha.github.io/virga/) and gCMCRT[109] (https://github.com/ELeeAstro/gCMCRT).

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

**Acknowledgements** This work is based on observations made with the NASA/ESA/CSA JWST. The data were obtained from the Mikulski Archive for Space Telescopes at the Space Telescope Science Institute, which is operated by the Association of Universities for Research in Astronomy, Inc., under NASA contract NAS 5-03127 for the JWST. These observations are associated with programme JWST-ERS-01366. Support for programme JWST-ERS-01366 was provided by NASA through a grant from the Space Telescope Science Institute, which is operated by the Association of Universities for Research in Astronomy, Inc., under NASA contract NAS 5-03127. L.A. acknowledges funding from STFC grant ST/W507337/1 and from the University of Bristol School of Physics PhD Scholarship Fund.

**Author contributions** All authors played a substantial role in one or more of the following: development of the original proposal, management of the project, definition of the target list and observation plan, analysis of the data, theoretical modelling and preparation of this manuscript. Some specific contributions are listed as follows. N.M.B., J.L.B. and K.B.S. provided overall programme leadership and management. L.A. and H.R.W. led the efforts for this manuscript. D.K.S., E.M.-R.K., H.R.W., I.J.M.C., J.L.B., K.B.S., L.K., M.L.-M., M.R.L., N.M.B., V.P. and Z.K.B.-T. made notable contributions to the design of the programme. K.B.S. generated the observing plan, with input from the team. E.S. and N.E. provided instrument expertise. B.B., E.M.-R.K., H.R.W., I.J.M.C., J.L.B., L.K., M.L.-M., M.R.L., N.M.B. and Z.K.B.-T. led or co-led working groups and/or contributed to important strategic planning efforts, such as the design and implementation of the prelaunch Data Challenges. A.L.C., D.K.S., E.S., N.E., N.P.G. and V.P. generated simulated data for prelaunch testing of methods. L.A., H.R.W., N.E.B. and J.D.L. contributed substantially to the writing of this manuscript, along with contributions in Methods from J.A.R., S.B., M.D., N.E., L.F., J.M.G., D.G., J.I., T.M.-E., P.-A.R. and N.L.W. L.A., H.R.W., M.K.A., J.A.R., S.B., M.D., N.E., L.F., D.G., J.I., T.M.-E., P.-A.R. and N.L.W. contributed to the development of data-analysis pipelines and/or provided the data-analysis products used in this analysis, that is, reduced the data, modelled the light curves and/or produced the planetary spectrum, with further contributions from J.Brande, T.D. and L.R.-R. J.D.L., N.E.B., J.M.G., E.K.H.L. and R.H. generated theoretical model grids for comparison with data. H.R.W., J.D.L. and N.E.B. generated figures for this manuscript. M.L.-M., K.D.C., N.P.G., L.K., M.L., J.I.M. and E.S. provided substantial feedback to the manuscript, coordinating comments from all other authors. T.D. is a LSSTC Catalyst Fellow, N.H.A. and A.D.F. are NSF Graduate Research Fellows, J.Kirk is an Imperial College Research Fellow, R.J.M., M.M., D.P., J.D.T. and L.W. are NHFP Sagan Fellows and B.V.R. is a 51 Pegasi b Fellow.

**Competing interests** The authors declare no competing interests.

**Additional information**
**Correspondence and requests for materials** should be addressed to Lili Alderson or Hannah R. Wakeford.

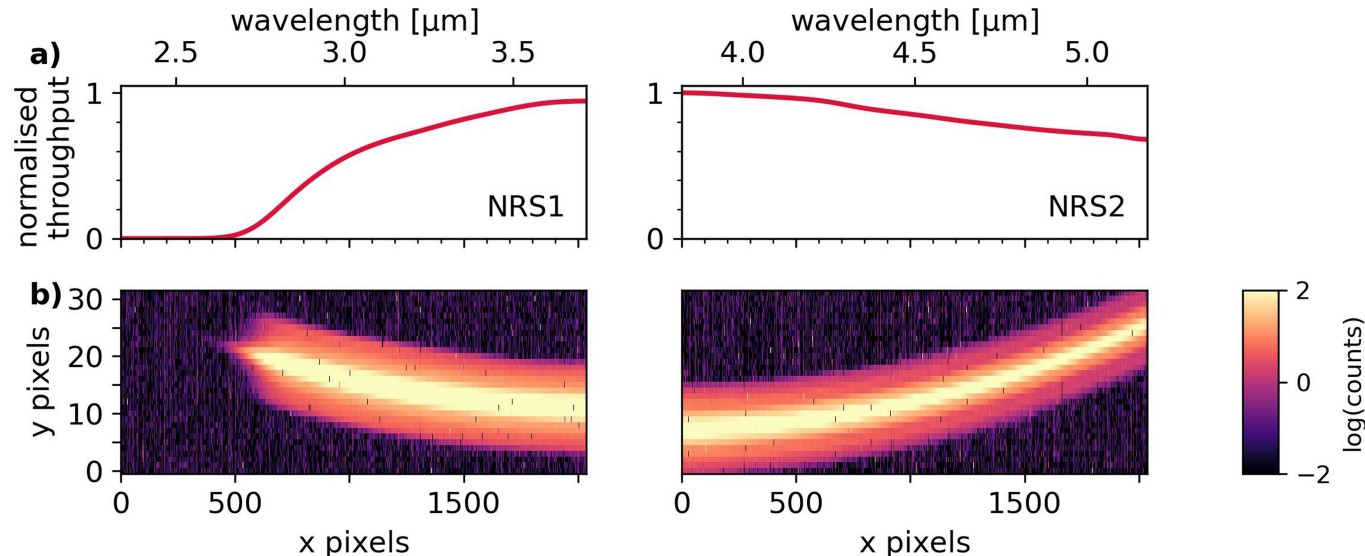

**Extended Data Fig. 1 | The throughput and spectral trace for WASP-39 across NRS1 and NRS2. a**, Normalized throughput of NRS1 and NRS2 detectors (as custom produced; see Methods, 'Limb-darkening'), which shows the cutoff at short wavelengths. **b**, 2D spectral images of the trace produced from the ExoTiC-JEDI [V1] reduction before cleaning steps. The aspect ratio has been stretched in the *y* direction to show the structure of the trace over the 32-pixel-wide subarray more clearly. The NRS2 spectral position is slightly offset from that of NRS1, as the NRS2 subarray was moved following commissioning to ensure that the centre of the spectral trace fell fully on the detector and did not fall off the top-right corner[139].

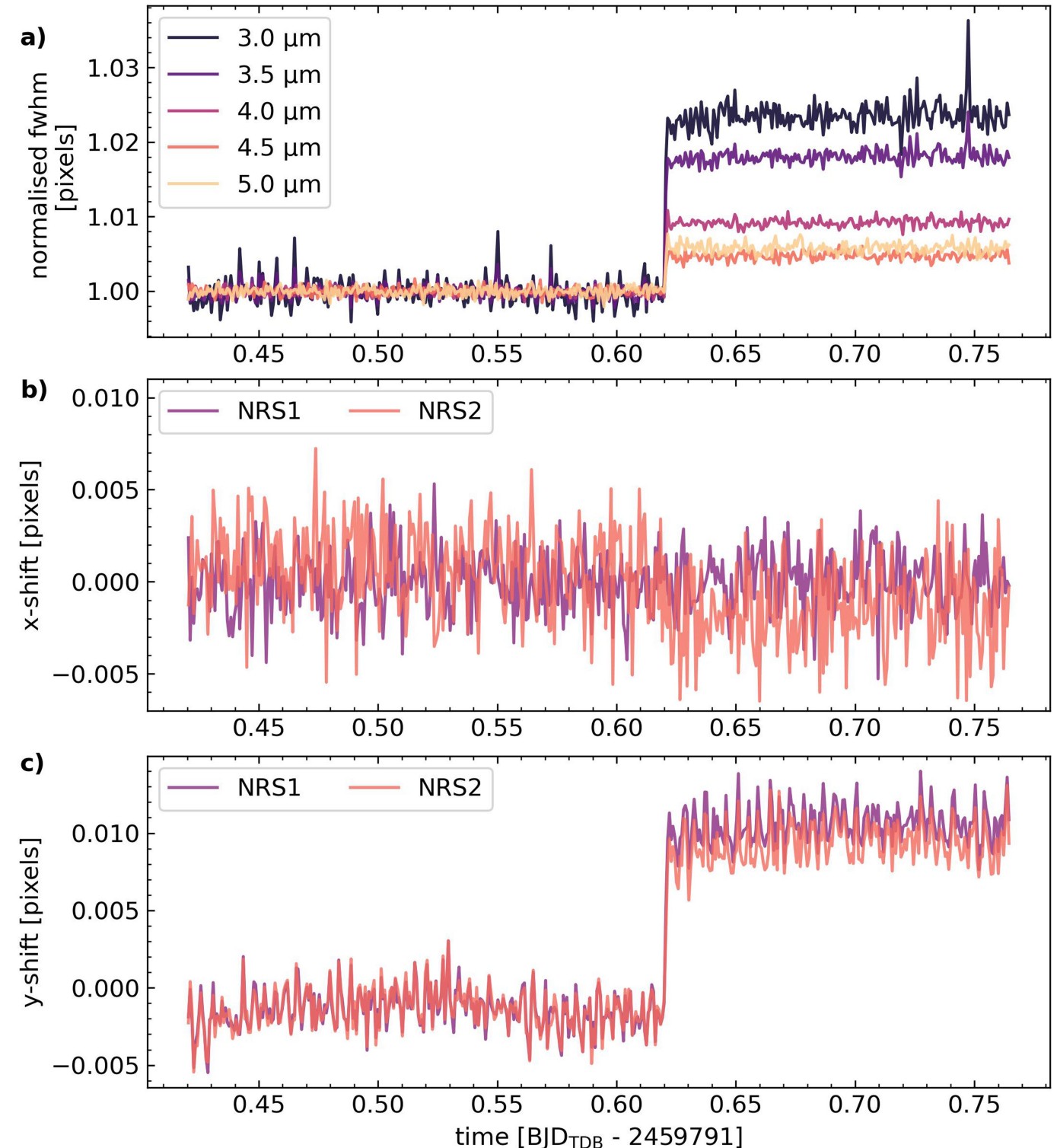

**Extended Data Fig. 2 | Time-dependent decorrelation parameters.**
**a**, The change in the FWHM of the spectral trace at selected wavelengths. This change does not correspond to any high-gain antenna movements and is attributed to a large mirror-tilt event. These measurements demonstrate that the mirror-tilt event has a wavelength dependence. Changes to the PSF have a larger impact at short wavelengths, as the PSF of the spectrum increases with wavelength[139]. **b,c**, The change in the $x$-pixel and $y$-pixel position of the spectral trace as functions of time, respectively. Positional shifts are calculated by cross-correlating the spectral trace with a template to measure sub-pixel movement on the detector. The $y$-position shift clearly shows a link to the mirror-tilt event.

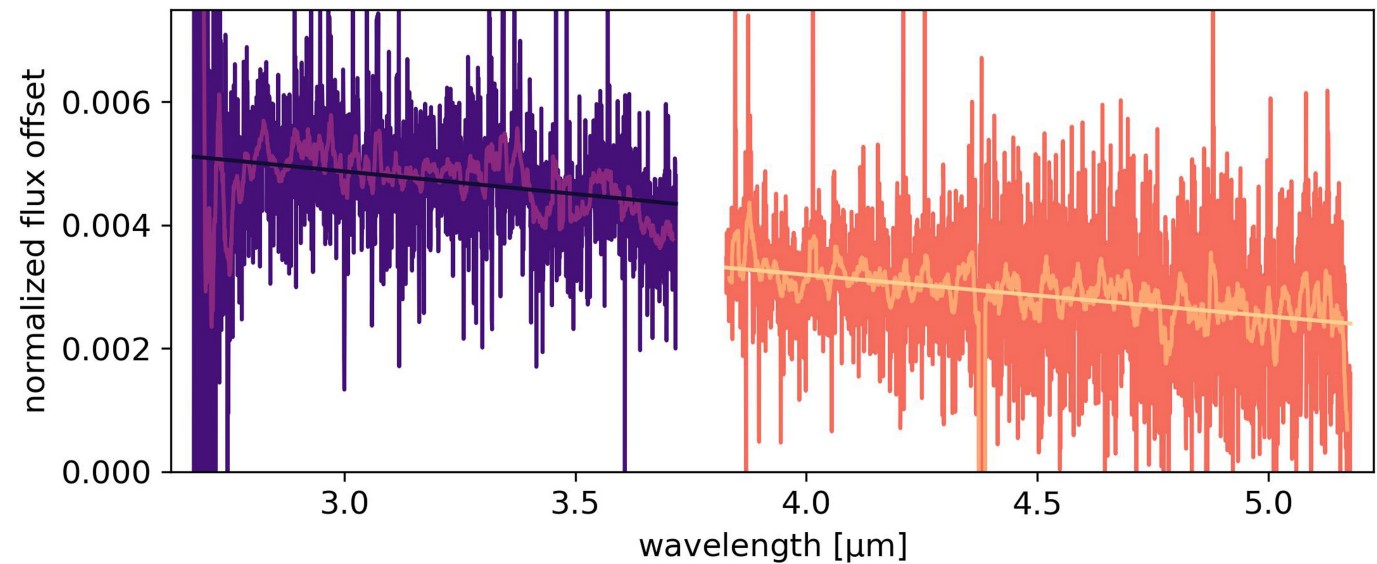

**Extended Data Fig. 3 | Normalized flux offset of the stellar baseline before and after the tilt event as a function of wavelength for NRS1 and NRS2.** Purple denotes NRS1 and orange denotes NRS2. The normalized flux offset is calculated per pixel by measuring the median flux in the stellar baseline before and after the transit and calculating the difference. These differences are then normalized by the before-transit flux and plotted on a common scale. Overplotted are the data binned to a resolution of 10 pixels to match our presented transmission spectra (Fig. 2). We also show a linear fit to each detector to better quantify the decreasing tilt flux amplitude with increasing wavelength (NRS1 = $-0.00073374x + 0.00707344$, NRS2 = $-0.00067165x + 0.00588128$).

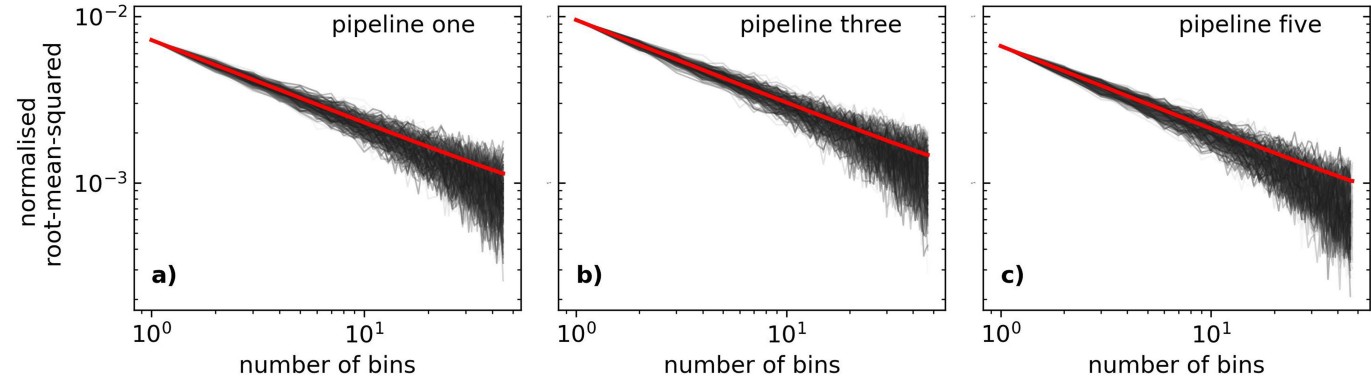

**Extended Data Fig. 4 | Normalized root-mean-squared binning statistic for three of the 11 reductions detailed in Methods.** In each subplot, the red line shows the expected relationship for perfect Gaussian white noise. The black lines show the observed noise from each spectroscopic light curve for pipelines 1, 3 and 5. To compare bins and noise levels, values for all bins in each pipeline are normalized by dividing by the value for a bin width of 1.

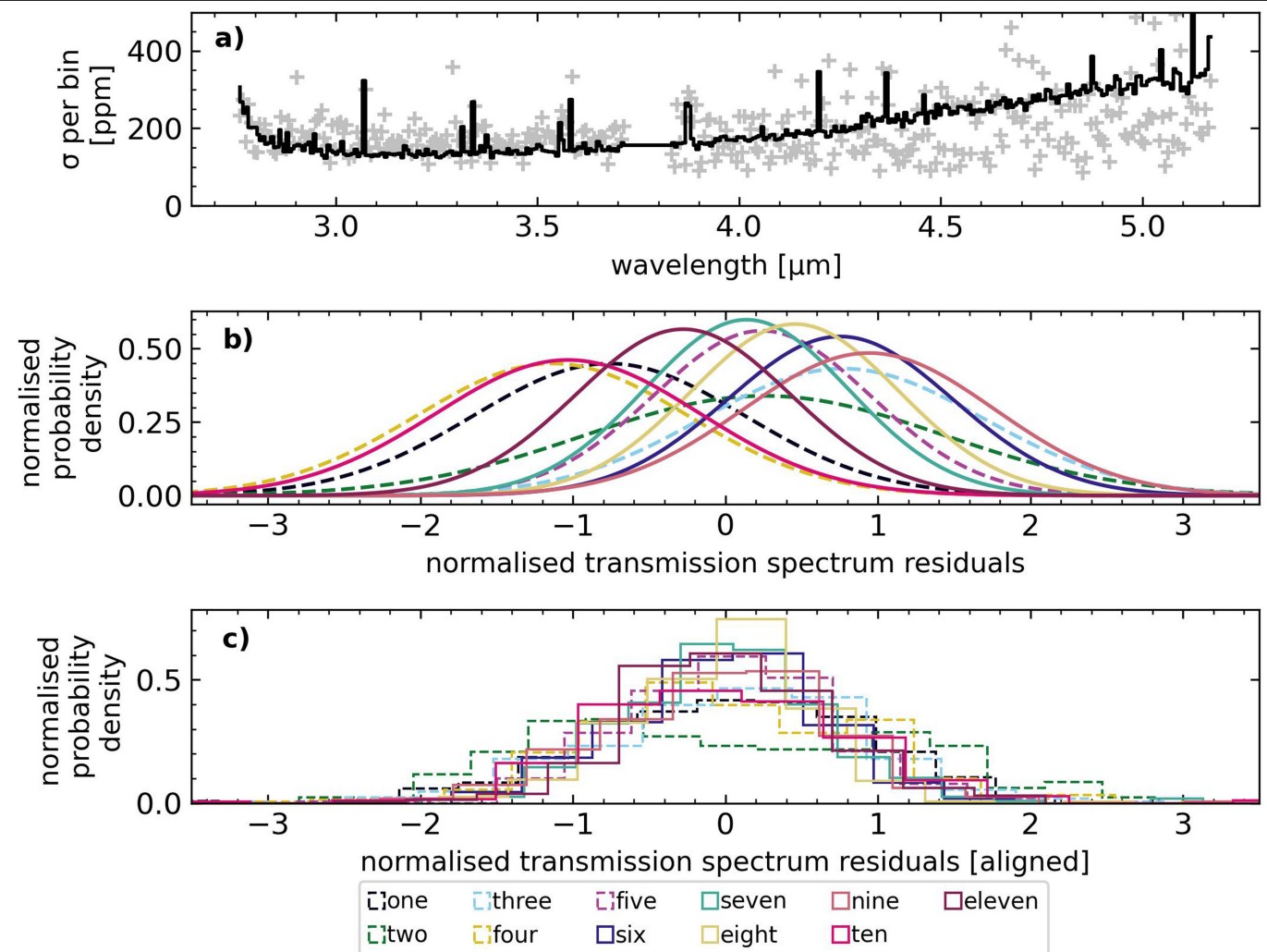

**Extended Data Fig. 5 | Comparison between all fitting pipelines performed on the spectroscopic light curves. a**, The underlying grey data points show the standard deviation between all transmission spectra per spectral bin. The black line shows the unweighted mean uncertainty on the transit depth per bin. Spikes in the uncertainties correspond to spectral bins with higher standard deviations, probably because of differences in pixel-flagging or sigma-clipping at the light-curve level. **b**, Gaussian PDFs of the normalized transmission spectrum residuals, showing the mean offset and the spread relative to the weighted-average transmission spectrum. **c**, Histograms of the normalized transmission spectrum residuals aligned to zero by subtracting the mean of the distribution that was used to generate the PDF above. In panels **b** and **c**, the coloured lines and numbers correspond to the fitting pipeline used to obtain each transmission spectrum, as summarized in Extended Data Table 1. The dashed lines correspond to the fitting pipeline results presented in Fig. 2, demonstrating that they are drawn from across the distribution.

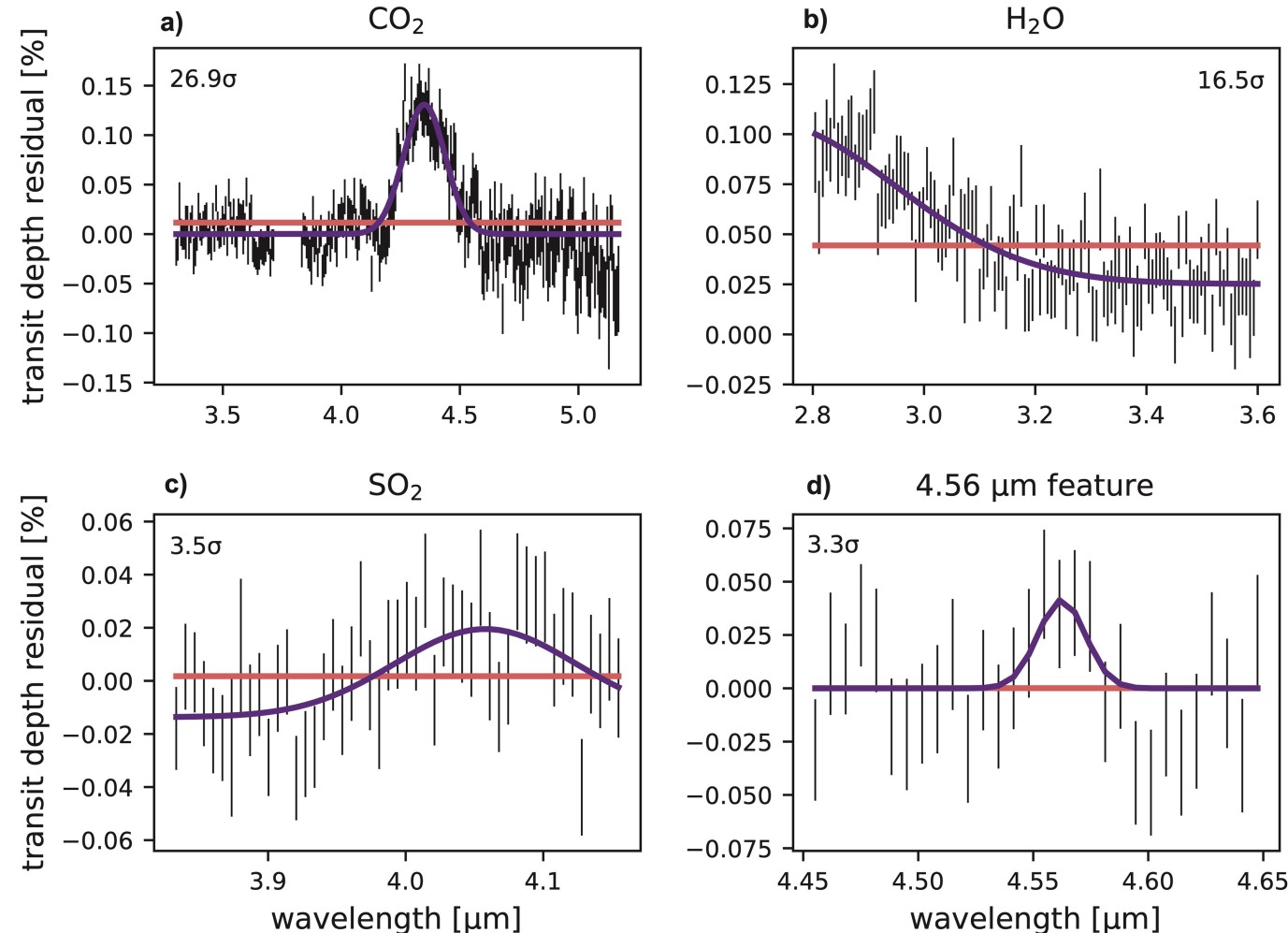

**Extended Data Fig. 6 | Gaussian versus flat-line fits to the residual transmission spectrum for $CO_2$, $H_2O$, $SO_2$ and the 4.56-µm feature.** Shown after all other absorption from the best-fit model is subtracted from the data. Each of the Gaussian fits has a higher Bayesian evidence than the flat-line fits, indicating a detection, although to varying degrees of significance.

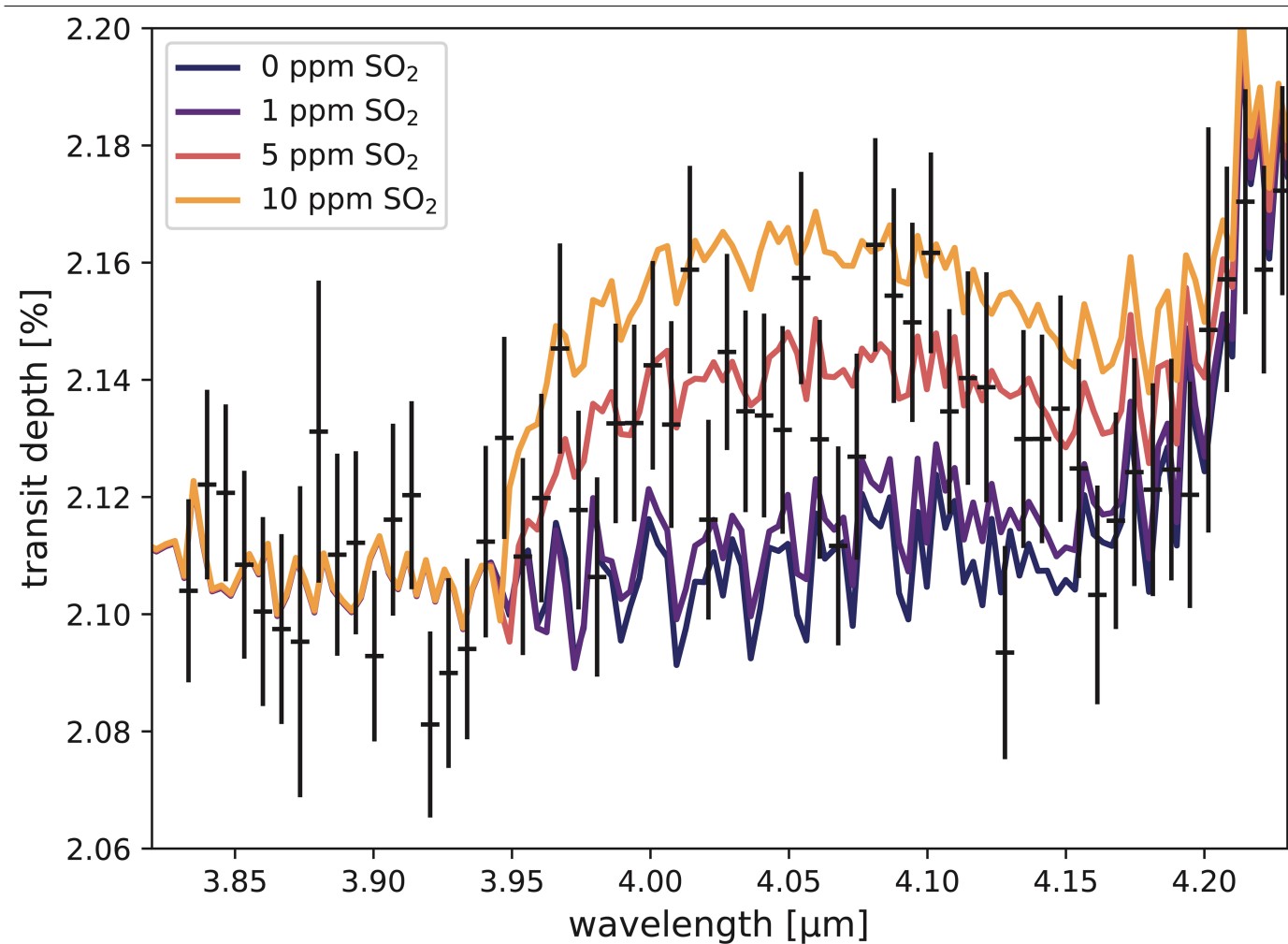

**Extended Data Fig. 7 | Model transmission spectra of WASP-39b with PHOENIX and gCMCRT with varying abundances of SO₂.** Model transmission spectra compared with the observed spectral feature at 4.1 μm in the G395H data. At wavelengths short of 3.95 μm, which is outside the SO₂ band, all models overlap, further suggesting that the data can be explained by the presence of SO₂ in the atmosphere. By interpolating these 10 times solar metallicity models, we find a best-fit SO₂ abundance of 4.6 ± 0.67 ppm. With the best-fit PICASO 3.0 at 3 times solar metallicity, we find an SO₂ abundance of 2.5 ± 0.65 ppm.

**Extended Data Table 1 | Summary of transit light-curve fitting**

| Transmission spectrum | Spectral images | Stellar spectrum | Fitting process | Limb-darkening |
|---|---|---|---|---|
| 1 | custom bias subtracted group-level destriped | ExoTiC-JEDI [V3] | LM, x, y, tilt normalised | non-linear, fixed |
| 2 | custom group-level destriped | Tiberius | GP [squared exponential kernel] + MCMC | quadratic, fit u1, fixed u2 |
| 3 | custom group-level destriped | ExoTiC-JEDI [V1] | LM, x, y, linear regression | non-linear, fixed |
| 4 | default rateints | transitspectroscopy | Nested sampling, linear(t), step(t) | square-root, fit |
| 5 | custom group-level destriped | Eureka! [V1] | MCMC, linear(t), constant tilt step | quadratic, fit u1 & u2 |
| 6 | custom group-level destriped | ExoTiC-JEDI [V1] | LM, x, y, exponential ramp, tilt decorrelated against y-shifts | non-linear, fixed |
| 7 | default rateints | Eureka! [V2] | MCMC, quadratic(t), tilt normalised | non-linear (ExoTHETyS), fixed |
| 8 | custom group-level destriped | ExoTiC-JEDI [V1] | MCMC, x, y | quadratic, fixed |
| 9 | custom group-level destriped | ExoTiC-JEDI [V1] | MCMC, x, y | quadratic, fixed |
| 10 | custom group-level destriped | ExoTiC-JEDI [V2] | GP [time-dependent Matern kernel], linear(t), sigmoid function for tilt event | quadratic, fixed |
| 11 | default rateints | Eureka! [V2] | MCMC, linear(t), step(t) | quadratic, fit u1 & u2 |

MCMC - Markov chain Monte Carlo, LM - least-squares minimiser, GP - gaussian process

An outline of the combined products and fitting pipelines used to compute each transmission spectrum.

| Grid Name | ATMO | PHOENIX | PICASO+Virga |
|---|---|---|---|
| Resolution/Sampling | R = 1000 & R = 3000 (Corr-k) | 1 Å sampling | R=60,000 (resampling) |
| Wavelength Range | 0.2 - 30 μm | 0.2-5.35 μm | 0.3-14 μm |
| **Global parameters** | | | |
| Internal temperature (K) | **100**, 200, 300, 400 | 200, **400** | **100**, 200, 300 |
| Heat redistribution | f = 0.25, **0.5**. 0.75, 1.0 (0.5=full, 1 = no redistribution) | f = 0.172, 0.25, **0.351** (0.25 = full redistribution) | f = 0.4, **0.5** (0.5=full redistribution) |
| **Chemistry parameters** | | | |
| Metallicity | 0.1, 1, **3**, 5, 10, 50, 100, 200× solar | 0.1, 1, **10**, 50, 100× solar | 0.1, 0.3, 1, **3**, 10, 30, 50, 100× solar |
| C/O ratio | **0.35**, 0.55, 0.7, 0.75, 1.0, 1.5 | **0.3**, 0.54, 0.7, 1.0 | 0.229, **0.458**, 0.687, 0.916 |
| Elemental Abundance Reference | Ref [141,142] | Ref [142] | Ref [143] |
| Solar C/O | 0.55 | 0.54 | 0.458 |
| **Aerosol parameters** | | | |
| $f_{sed}$ | N/A | N/A | **0.6**, 1, 3, 6, 10 |
| $K_{zz}$ (cm²/s) | N/A | N/A | 1e5, 1e7, **1e9**, 1e11 |
| Cloud Opacities | Grey (kappa factor - 0.0, 0.5, 1.0, **5.0**) | Grey | MnS, $Na_2S$, $MgSiO_3$, grey |
| $P_{cloud}$ | **1** to 50 mbar (fixed) | None, **0.3**, 1, 3, 10 mbar | Variable (fit on the fly) |
| Rayleigh scattering | **$H_2$ only**, 10× | **$H_2$ only**, 10× | $H_2$ only |
| **Molecular and Atomic Opacity Sources Included** | | | |
| | $CH_4$, CO, $CO_2$, $C_2H_2$, Cs, FeH, HCN, $H_2O$, $H_2S$, K, Li, Na, $NH_3$, $PH_3$, Rb, $SO_2$, TiO, VO | CH, $CH_4$, CN, CO, $CO_2$, COF, $C_2$, $C_2H_2$, $C_2H_4$, $C_2H_6$, CaH, CrH, FeH, HCN, HCl, HF, HI, HDO, $HO_2$, $H_2$, $H_2S$, $H_2O$, $H_2O_2$, $H_3+$, MgH, NH, $NH_3$, NO, $N_2$, $N_2O$, OH, $O_2$, $O_3$, $PH_3$, $SF_6$, SiH, SiO, $SiO_2$, TiH, TiO, VO, atoms up to U | $CH_4$, CO, $CO_2$, $C_2H_2$, $C_2H_4$, $C_2H_6$, CrH, Cs, Fe, FeH, HCN, $H_2$, $H_2O$, $H_2S$, $H_3+$, K, Li, LiCl, LiH, MgH, $NH_3$, $N_2$, Na, OCS, $PH_3$, Rb, SiO, TiO, VO |
| **Statistical Parameters for best-fit (bolded) model** | | | |
| $\chi^2$/N (N=344) | 1.098 | 1.203 | 1.084 |

The parameter space explored by each RCTE model grid. The best-fit model for each grid is shown in bold. Ref. [140–142].

**Extended Data Table 3 | Detection significances**

| Gas | Bayesian Gas Removal | | Gaussian Residual Fit |
|-----|---------|------|------|
|     | ln $B$  | σ    | σ    |
| $H_2O$ | 402.6 | 21.5 | 16.5 |
| $CO_2$ | 229.0 | 28.5 | 26.9 |
| $SO_2$ | 9.7 | 4.8 | 3.5 |
| CO | -5.0 | 0.3 | 4.5 |

Feature-detection significance for dominant sources of opacity with two different methods. *B* is the Bayes factor.