## [Peer Review File · Nature]

Manuscript Title: Early Release Science of the Exoplanet WASP-39b with JWST NIRSpec G395H

Reviewer Comments & Author Rebuttals

Reviewer Reports on the Initial Version:

Referees' comments:

Referee #1 (Remarks to the Author):

I would like to congratulate the authors on a well-written manuscript in a very short time frame since the data was obtained. Please find my review comments in the following:

A. The authors present NIRSpec observations with JWST from the Early Release Science of the exoplanet WASP-39b, a well-studied exoplanet. In this work, they achieve very significant detections of CO₂ and H₂O, two important tracers for the calculation of C/O ratios. C/O ratios in turn inform planet formation theories and in this case underscore the importance of photochemistry in exoplanet atmospheres. They also present the identification of SO₂ as the source of a previously unknown feature at 4.1 micro m.

B. The authors themselves acknowledge that the detection of both CO₂ and H₂O are not novel but both species have been regularly studied from space with Hubble and Spitzer (see references 3-5 in the manuscript) and from the ground (see references 6-8 in the manuscript). From my point of view, the only novelty presented in this manuscript is the use of JWST and the subsequent increase in precision. This kind of analysis and paper will be one of the standard results coming from JWST in the years to come. In my opinion, this paper, while well-written and definitely worthy of publication does not provide any novel insights into planetary C/O ratios or species detection to the community. I leave the decision of whether the use of JWST is an immediate justification for publication in Nature to the editor.

C. and D. and E. I am not an expert in space-based observations and will leave any comments on the technical aspects of JWST and the data preparation to the other referees. In the following I will provide pertinent comments chronologically:

In line 104 the authors state that their analysis of different fitting and reduction methods results "in transmission spectra consistent within 2.95 sigmas". That seems like an extraordinarily wide spread between the different approaches when one aims normally for consistency within 1 sigma. Could the authors please provide more clarity here as to why the difference between the approaches is so large (see also the spread in Extended Data Figure 5? What is the impact of this deviation on the conclusions?

In line 122 ff, the authors mention two unknown spectral features one at 4.56 micro m and one at 4.1 micro m subsequently attributed to SO₂. They also mention "an exhaustive search" for possible

opacity sources. Please provide more information on this point. What lines/bands were looked at? What are the possible lines/bands in both regions from the NIST database? Which molecules could have rotational or vibrational excitation states at these wavelengths? Which ions could produce these features etc? As to SO₂, it is unclear why it is the only candidate. Which lines/bands were identified and from which sources? Are there any alternative explanations for this feature?

In line 506 ff the impact of limb darkening is assessed thoroughly. Does the Rossiter-McLaughlin effect play any role in your analysis?

In line 914, the authors state that clouds were incorporated in their models using PyMultiNest to test for feature significance. For the other features, only Chi-square assessments are mentioned. Why was a more robust retrieval approach using PyMultiNest not employed throughout? This would provide much better coverage of the parameter space.

G. The most crucial references in this work point to other JWST papers either submitted or accepted in Nature (e.g. Ahrer+Submitted, Rustamkulov+Submitted, ERS+accepted). I cannot speak about whether appropriate credit was given here, or whether the references are appropriate given that I do not have access to any of these publications. I am reluctant to endorse a paper that leans on work that is not currently accessible to the community and in two cases has not yet passed peer review. I would urge the authors to provide more details on the steps where the mentioned papers are cited to make this manuscript a more standalone work.

H. The paper is easily understandable and appropriate to the subject. I have found it a pleasure to read.

I would like to congratulate the authors again on their exhaustive work and wish them much success with future JWST datasets.

Referee #2 (Remarks to the Author):

I highly appreciate the effort made by the authors to report results of JWST early release science. This article reports transit spectroscopy for WASP-39 b using NIRSpec G395H on JWST. The authors derived a set of transmission spectra of WASP-39b using various analysis pipelines, and made comparisons with 3 different atmospheric models to characterize atmospheric compositions of WASP-39 b. The authors claim significant detections of CO₂, H₂O, and SO₂. This article demonstrates the performance of JWST NIRSpec G395H from the JWST early release science. The results and analyses presented in this article are of great interest to the community. Thus I believe this article is eventually worth publishing in Nature. However, I would suggest a moderate revision before publishing.

I have a concern about the claim of the SO₂ detection. I can agree that the SO₂ is a possible opacity source, but I am still not convinced of the SO₂ detection from the current manuscript. If the authors hope to present the SO₂ detection as a major finding of this article, more convincing explanations are desired. For this reason, I hope the authors address the following questions to prove the

robustness of the SO₂ detection.

(1) According to Extended Data Table 2, initially only the ATMO model includes SO₂ as an opacity source, and the PHOENIX and the PICASO+Virga models do not. Subsequently the authors presented analyses for feature detection and SO₂ detection from line 927 to 994. But the authors do not report any result on the SO₂ detection from the ATMO model. What was the result of the SO₂ absorption for the ATMO model?

(2) As noted above, Extended Data Table 2 indicates that the PICASO+Virga model does not include SO₂ as an optically source. While, the text (lines 899-900) indicates SO₂ is included as an opacity source. This is confusing and contradictory. Which is correct information?

(3) If the text is correct and SO₂ was included as an opacity source from the beginning, why was the best-fit model (reduced $\chi^2 = 1.02$) mentioned in lines 979-980 not found initially?

(4) If Extended Data Table 2 is correct and SO₂ was not included as an opacity source initially, why are the PICASO best-fit models presented in lines 910-912 and lines 975-976 so different?

(5) Lines 928-932 introduce all potential opacity sources including minor ones, but SO₂ is not listed here. Then SO₂ suddenly appears in line 933. Is this intentional or is there any typo?

(6) In line 990, the authors report the reduced χ^2 between 3.8 - 4.2 μm for the PHOENIX+SO₂ model is 0.80. What was the reduced χ^2 for the whole wavelength, compared with the original result presented in Extended Data Table 2 (i.e., 1.203)?

Followings are other relatively minor comments.

- line 20-21

σ first appears here. But the notation of σ is only defined in line 794 in this article. I suppose the meanings of σ in line 20-21 and 794 (and others such as in line 369) are different. To avoid confusion, the authors should describe the definition of this σ explicitly here. Also, it is better to avoid using an identical notation for different meanings without explanations.

- line 21

Although the authors describe "Best-fit atmosphere models range between 3x and 10x solar metallicity," according to Extended Data Table 2, 10x solar metallicity is preferred for the PHOENIX model, but the PHOENIX model does not have any grid between 1x - 10x solar metallicity. So the result of 10x solar metallicity appears to be mainly due to the absence of the grids, and the best estimate can be lower than 10x solar metallicity and is just unknown due to the lack of the grids between 1x - 10x solar. I feel odd here as this fact is not presented in the main text. To avoid misunderstanding, how about adding "Among available parameter grids," at the beginning of the sentence?

- line 22

What is the definition of the solar C/O ratio in this article? It appears the authors define 0.458 (used

in the PICASO+Virga model, mentioned in line 911) as the solar C/O ratio in this article. But the definition conflicts with the commonly used value of about 0.55 (e.g., adopted in a previous Nature paper, <https://arxiv.org/abs/2110.14821>). The authors should define the value of the solar C/O ratio in this article clearly in the main text and present a reference.

- line 136

Reduced χ^2 (χ^2/N) first appears here. But no definition of N is given in the main text (first presented in line 883). The authors should present the definition and the value of N here. Also, the reduced χ^2 is described in two different ways in this article: χ^2/N (e.g., line 136) and χ^2_{ν} (e.g., line 976). The notation should be unified.

- Figure 3

To me, it appears the PHOENIX model is almost equivalently good fit from the lower panel. But the reduced χ^2 is very different. I suppose the difference of the atmospheric models around 4.3-4.4 μm in the upper panel is the cause of the difference of the reduced χ^2 . But it appears the χ^2 difference in the wavelength range is almost negligible according to the lower panel. I would like the authors to specify and describe which wavelength range makes the PHOENIX model worse than other 2 models.

- line 336-337

Similar values for the changes of the reduced χ^2 for the ATMO and PHOENIX models should be presented to strengthen the claim of the SO₂ detection.

- line 369

Why was the threshold of 15σ adopted opposed to the default of 4σ ? For the sake of future observers, please give a brief reason. What is the impact when adopting a different threshold?

- line 404-405

Similarly, please give a brief reasoning of the thresholds of 6σ and 20σ , and the impact of different thresholds.

- line 430-434

I could not understand the sentences. There are 2 identical sentences here, and it is very confusing. Again, please give a brief reasoning of the thresholds of 5σ and 3σ , and the impact of different thresholds.

- line 447

The authors describe "3 standard deviations" here. Is this meaning different from 3σ (e.g., in line 442)? Why is the different notation used here?

- line 488-489

Please give a brief reasoning of the thresholds of 10σ and 3σ , and the impact of different thresholds.

- line 506

Should be "Limb-darkening" according to the following paragraph.

- the section of "Light Curve Fitting"

I highly appreciate the authors' effort for testing those different pipelines and presenting comparisons of resultant transmission spectra. I would further appreciate it if the authors would present useful guidelines for future observers here. Namely, what is the authors' recommendation for future observers based on the authors' experience from the early release science? Do the authors recommend doing a similar analysis (i.e., using 11 pipelines and combined) for future observers? Or is any pipeline fine? Or which pipeline is recommended? Such information would be extremely useful for future observers and worth reporting as a result of the early release science.

- Extended Data Table 1, line 854

The abbreviation of "LD - limb-darkening" is defined here, but not used in the Table. Is this necessary?

- the section of "Model Comparison"

The order of 3 models are different in the text (PHOENIX, ATMO, PICASO+Virga) and Extended Data Table 2 (ATMO, PHOENIX, PICASO+Virga).

- Extended Data Table 3, line 924

Although the authors describe "B is the Bayesian evidence," it appears B is the Bayes factor between the models including and excluding a specific opacity source. Is this a typo, or do the authors use the same meaning for the Bayesian evidence and the Bayes factor?

- line 1004

I understand the 4.8σ detection of SO₂ mentioned in the main text is statistical (Bayesian) significance of the presence of SO₂, and here the authors present the amount of SO₂ based on the Levenberg-Marquardt regression. But apparently the description of SO₂ abundance of 2.5 ± 0.65 ppm does not match 4.8σ detection. To avoid any confusion by readers, please give an explanation on what is the definition of the 1σ uncertainty of 0.65 ppm here.

Author Rebuttals to Initial Comments:

Referees' comments:

Referee #1 (Remarks to the Author):

I would like to congratulate the authors on a well-written manuscript in a very short time frame since the data was obtained. Please find my review comments in the following:

A. The authors present NIRSpec observations with JWST from the Early Release Science of the exoplanet WASP-39b, a well-studied exoplanet. In this work, they achieve very significant detections of CO₂ and H₂O, two important tracers for the calculation of C/O ratios. C/O ratios in turn inform planet formation theories and in this case underscore the importance of photochemistry in exoplanet atmospheres. They also present the identification of SO₂ as the source of a previously unknown feature at 4.1 micro m.

B. The authors themselves acknowledge that the detection of both CO₂ and H₂O are not novel but both species have been regularly studied from space with Hubble and Spitzer (see references 3-5 in the manuscript) and from the ground (see references 6-8 in the manuscript). From my point of view, the only novelty presented in this manuscript is the use of JWST and the subsequent increase in precision. This kind of analysis and paper will be one of the standard results coming from JWST in the years to come. In my opinion, this paper, while well-written and definitely worthy of publication, does not provide any novel insights into planetary C/O ratios or species detection to the community. I leave the decision of whether the use of JWST is an immediate justification for publication in Nature to the editor.

As the referee notes, H₂O and CO₂ have both been detected before, however, CO₂ detections were widely regarded as tentative while the measurement presented leaves little doubt as to its origin. In addition, we present the novel detection of photochemical products in this planet's atmosphere which have only been revealed with JWST data. The resolution and wavelength coverage of the G395H data presented in this manuscript is unique to this mode on JWST. As such, the data present a new look at what we can expect from exoplanet science in the future. The goal of the early release science with JWST is to demonstrate to the community the standard of the results possible with these instrument modes, in this paper, we present 11 reduction methods that can all replicate the results and discuss the implications with regard to data quality and planning which is needed at this time right at the start of decades of state of the art observations.

C. and D. and E. I am not an expert in space-based observations and will leave any comments on the technical aspects of JWST and the data preparation to the other referees. In the following I will provide pertinent comments chronologically:

In line 104 the authors state that their analysis of different fitting and reduction methods results "in transmission spectra consistent within 2.95 sigmas". That seems like an extraordinarily wide spread between the different approaches when one aims normally for consistency within 1 sigma. Could the authors please provide more clarity here as to why the difference between the approaches is so large (see also the spread in Extended Data Figure 5)? What is the impact of this deviation on the conclusions?

We thank the referee for raising this as it is an important aspect of our detailed analysis across multiple reduction methods. As stated in the paper, before accounting for offsets between the derived transmission spectra all transmission spectral points across all 11 pipelines fall within 2.95 sigma of the weighted average transmission spectrum. While this may at first look sound large, this is in fact remarkable given the wide range of methods used to measure the transmission spectrum. However, as highlighted by the referee's comment, is potentially not the best metric to describe the agreement across all reductions. We show in Extended Data Figure 5 that without accounting for absolute offsets, likely caused by limb-darkening procedures or data cleaning techniques (as described in Methods, Transmission Spectral Analysis section), the peak of the residuals for all transmission spectra lie within 1.2 sigma of the weighted average result but that the tails of the distributions extend out to the 2.95-sigma as quoted. ED Figure 5 c) then shows the overall agreement between the spectra once absolute offsets are taken into account. This shows that the structure, and thus absorption signatures, of the transmission spectra are consistent across all 11 reductions further backing up the results presented.

Due to space constraints in the main text this discussion has been moved to the Methods section and detailed as above.

In line 122 ff, the authors mention two unknown spectral features one at 4.56 micro m and one at 4.1 micro m subsequently attributed to SO₂. They also mention "an exhaustive search" for possible opacity sources. Please provide more information on this point. What lines/bands were looked at? What are the possible lines/bands in both regions from the NIST database? Which molecules could have rotational or vibrational excitation states at these wavelengths? Which ions could produce these features etc? As to SO₂, it is unclear why it is the only candidate. Which lines/bands were identified and from which sources? Are there any alternative explanations for this feature?

This paper will be presented in conjunction with a series of papers analysing the transmission spectrum of WASP-39b across four different instrument modes, as detailed in the main text. Across these papers, there is overlapping information that is shared across the series to avoid repetition of the same analysis. We now clearly cite the JWST TEC-ERS

paper analyzing the NIRSPEC/PRISM data where the search for plausible opacity sources is described. For reference, here is the relevant text from that paper:

“None of the 1D RCTE models are able to capture the 4 μ m absorption feature seen in the data. We searched for multiple candidate gas species that could produce this feature if their abundances differ from the expected abundances from thermochemical equilibrium. The list of searched chemical species include C-bearing gases like C₂H₂, CS, CS₂, C₂H₆, C₂H₄, CH₃, CH, C₂, CH₃Cl, CH₃F, CN, and CP. Various metal hydrides, bromides, fluorides and chlorides such as LiH, AlH, FeH, CrH, BeH, TiH, CaH, HBr, LiCl, HCl, HF, AlCl, NaF, and AlF were also searched as potential candidates to explain the feature. SO₂, SO₃, SO, and SH are among the sulphur-based gases which were considered. Other species which were considered include gases like PH₃, H₂S, HCN, N₂O, GeH₄, SiH₄, SiO, AsH₃, H₂CO, H⁺³, OH⁺, KOH, Br α -H, AlO, CN, CP, CaF, H₂O₂, H₃O⁺, HNO₃, KF, MgO, PN, PO, PS, SiH, SiO₂, SiS, TiO, and VO.

Among all these gases, SO₂ was the most promising candidate in terms of its spectral shape and chemical plausibility, although the expected chemical equilibrium abundance of SO₂ is too low to produce the absorption signal seen in the data. However, previous work exploring photochemistry in exoplanetary atmospheres [Tsai et al. 2021, Zahnle et al. 2009] have shown that higher amounts of SO₂ can be created in the upper atmospheres of irradiated planets through photochemical processes.”

In line 506 ff the impact of limb darkening is assessed thoroughly. Does the Rossiter-McLaughlin effect play any role in your analysis?

We are not able to measure the Rossiter-McLaughlin effect on the stellar lines at the resolution of these observations.

In line 914, the authors state that clouds were incorporated in their models using PyMultiNest to test for feature significance. For the other features, only Chi-square assessments are mentioned. Why was a more robust retrieval approach using PyMultiNest not employed throughout? This would provide much better coverage of the parameter space.

We clarify that this PyMultiNest framework was in fact used for all the feature detection significance shown in Extended Table 3. Within this framework, the parameter space is constrained to the grid parameters (using a nearest neighbour technique). The exception is 1) the cloud top pressure corresponding to a grey cloud deck with infinite opacity, 2) a radius scaling to account for the unknown reference pressure, and 3) the abundance of SO₂, for the SO₂ injection test described in “SO₂ Absorption”.

G. The most crucial references in this work point to other JWST papers either submitted or accepted in Nature (e.g. Ahrer+Submitted, Rustamkulov+Submitted, ERS+accepted). I cannot speak about whether appropriate credit was given here, or whether the references are appropriate given that I do not have access to any of these publications. I am reluctant to endorse a paper that leans on work that is not currently accessible to the community and in two cases has not yet passed peer review. I would urge the authors to provide more details on the steps where the mentioned papers are cited to make this manuscript a more standalone work.

We note that the papers referenced here are part of a joint publication strategy where all papers will be made available to the community at the same time. While the other papers detail associated observations with other instruments, the analysis and results presented do not rely on work presented in those papers and thus this paper is able to stand alone.

H. The paper is easily understandable and appropriate to the subject. I have found it a pleasure to read.

I would like to congratulate the authors again on their exhaustive work and wish them much success with future JWST datasets.

The authors would like to thank the referee for their kind words and for actively conveying this in the review process which is often taken as a critical exercise. We hope referees continue this practice and continue to see more positive responses to manuscripts in future published reviews.

Referee #2 (Remarks to the Author):

I highly appreciate the effort made by the authors to report results of JWST early release science. This article reports transit spectroscopy for WASP-39 b using NIRSpec G395H on JWST. The authors derived a set of transmission spectra of WASP-39b using various analysis pipelines, and made comparisons with 3 different atmospheric models to characterize atmospheric compositions of WASP-39 b. The authors claim significant detections of CO₂, H₂O, and SO₂. This article demonstrates the performance of JWST NIRSpec G395H from the JWST early release science. The results and analyses presented in this article are of great interest to the community. Thus I believe this article is eventually worth publishing in Nature. However, I would suggest a moderate revision before publishing.

I have a concern about the claim of the SO₂ detection. I can agree that the SO₂ is a possible opacity source, but I am still not convinced of the SO₂ detection from the current manuscript. If the authors hope to present the SO₂ detection as a major finding of this article, more convincing explanations are desired. For this reason, I hope the authors address the following questions to prove the robustness of the SO₂ detection.

The authors thank the reviewer for their critical insight into the detections presented. We note that a detailed paper on the mechanism behind SO₂ production will be presented in a separate manuscript as part of the wider JTEC ERS collaboration.

(1) According to Extended Data Table 2, initially only the ATMO model includes SO₂ as an opacity source, and the PHOENIX and the PICASO+Virga models do not. Subsequently the authors presented analyses for feature detection and SO₂ detection from line 927 to 994. But the authors do not report any result on the SO₂ detection from the ATMO model. What was the result of the SO₂ absorption for the ATMO model?

While SO₂ was indeed included in the ATMO models as an opacity source from the beginning, these models were all in radiative-convective equilibrium. As such, the abundance of the SO₂ in the ATMO model was at the equilibrium volume mixing ratio, which is less than 1e-10 and much too small to account for the absorption feature seen. This is why, even when including SO₂, the ATMO spectra are very similar to the PICASO and PHOENIX models in Figure 3. Only after increasing the SO₂ abundance do the models begin to fit the observed feature.

(2) As noted above, Extended Data Table 2 indicates that the PICASO+Virga model does not include SO₂ as an optically source. While, the text (lines 899-900) indicates SO₂ is included as an opacity source. This is confusing and contradictory. Which is correct information?

Extended Data Table 2 indicates the details of the original radiative-convective equilibrium model grids, which for PICASO+Virga and PHOENIX, did not include SO₂. Subsequent text describes the process whereby we included and adjusted the SO₂ abundance to these models to improve the fit from the original grids.

(3) If the text is correct and SO₂ was included as an opacity source from the beginning, why was the best-fit model (reduced $\chi^2 = 1.02$) mentioned in lines 979-980 not found initially?

As mentioned in the response to point (1), the best-fit model mentioned on lines 979-980 was found by increasing the abundance of SO₂ after it was realised that the shape and location of SO₂'s absorption matched that observed feature. Physically, this increased abundance of SO₂ is due to the molecule being a photochemical product, as mentioned in the text and the submitted publication Tsai et al. SO₂ as a significant photochemical product has

also been explored previously in Tsai et al. 2021 and Zahnle et al 2009 (both referenced in the text).

(4) If Extended Data Table 2 is correct and SO₂ was not included as an opacity source initially, why are the PICASO best-fit models presented in lines 910-912 and lines 975-976 so different?

Extended Data Table 2 specifically describes the opacity sources included in the RCTE models. As described in the main text: “The absorption feature seen at 4.1 μm is also not seen in the RCTE model grids”. We removed SO₂ from the “notable line list” when describing the RCTE models in “PICASO 3.0 & Virga” section. Then, we clarify the injection technique in “SO₂ absorption”: “We add SO₂ opacity using the ExoMol linelist 105, without rerunning the RCTE model to compute a new climate profile.”

(5) Lines 928-932 introduce all potential opacity sources including minor ones, but SO₂ is not listed here. Then SO₂ suddenly appears in line 933. Is this intentional or is there any typo?

Lines 928-932 refer to the models in chemical equilibrium, for which SO₂ does not have significant absorption. Through the process described above and in the text of trying additional opacity sources to fit the 4 μm feature, evidence for the presence of SO₂ was found (though with an out-of-equilibrium abundance).

(6) In line 990, the authors report the reduced χ^2 between 3.8 - 4.2 μm for the PHOENIX+SO₂ model is 0.80. What was the reduced χ^2 for the whole wavelength, compared with the original result presented in Extended Data Table 2 (i.e., 1.203)?

Inserting the SO₂ fit into the best-fit PHOENIX model improves the χ^2/N from 1.20 to 1.08. We now quote this improvement, rather than the specific fit between 3.8-4.2 μm .

Followings are other relatively minor comments.

- line 20-21

σ first appears here. But the notation of σ is only defined in line 794 in this article. I suppose the meanings of σ in line 20-21 and 794 (and others such as in line 369) are different. To avoid confusion, the authors should describe the definition of this σ explicitly here. Also, it is better to avoid using an identical notation for different meanings without explanations.

As the referee points out we use sigma to define three different instances in this analysis, in lines 20-21 as the significance of detection based on the bayesian information, on

line 794 to describe the uncertainty on the transmission data points from each reduction, and in the methods such as 369 where it is used a standard deviation from the residuals. All of these are formally denoted as sigma values in mathematics and in the literature as they are essentially rooted in the same statistic. We have clarified the terms with each use, however, feel that changing any one of them would likely lead to further confusion as all are well-defined uses of the term sigma.

- line 21

Although the authors describe "Best-fit atmosphere models range between 3x and 10x solar metallicity," according to Extended Data Table 2, 10x solar metallicity is preferred for the PHOENIX model, but the PHOENIX model does not have any grid between 1x - 10x solar metallicity. So the result of 10x solar metallicity appears to be mainly due to the absence of the grids, and the best estimate can be lower than 10x solar metallicity and is just unknown due to the lack of the grids between 1x - 10x solar. I feel odd here as this fact is not presented in the main text. To avoid misunderstanding, how about adding "Among available parameter grids," at the beginning of the sentence?

We have added a clarification in the main text (second to last paragraph): "...ranging from 3–10x solar metallicity, given individual model grid's spacing"

- line 22

What is the definition of the solar C/O ratio in this article? It appears the authors define 0.458 (used in the PICASO+Virga model, mentioned in line 911) as the solar C/O ratio in this article. But the definition conflicts with the commonly used value of about 0.55 (e.g., adopted in a previous Nature paper, <https://arxiv.org/abs/2110.14821>). The authors should define the value of the solar C/O ratio in this article clearly in the main text and present a reference.

Our assumptions for elemental ratios and references are already provided in Extended Table 3 under "Elemental Abundance". We have added a row for Solar C/O to this table in order to add additional transparency. However, regardless of the definition, our overall conclusion from the main text that our models "indicate C/O ratios ranging from sub-solar to solar depending on the grid used" is not dependent on these definitions.

- line 136

Reduced χ^2 (χ^2/N) first appears here. But no definition of N is given in the main text (first presented in line 883). The authors should present the definition and the value of N here. Also, the reduced χ^2 is described in two different ways in this article: χ^2/N (e.g., line 136) and χ_v^2 (e.g., line 976). The notation should be unified.

χ^2/N is first shown and described in the paragraph above the section quoted: “The best-fit models from each grid provide a reduced chi-square per data point (χ^2/N) of...”.

The use of χ^2 is a typo and has been corrected to χ^2/N

- Figure 3: To me, it appears the PHOENIX model is almost equivalently good fit from the lower panel. But the reduced χ^2 is very different. I suppose the difference of the atmospheric models around 4.3-4.4 μm in the upper panel is the cause of the difference of the reduced χ^2 . But it appears the χ^2 difference in the wavelength range is almost negligible according to the lower panel. I would like the authors to specify and describe which wavelength range makes the PHOENIX model worse than other 2 models.

As you indicate, the difference on the blue edge of the CO₂ absorption feature appears to be the cause of the difference in χ^2 . This is due to a difference in linelists used. We now indicate this in the description of the PHOENIX grid where the difference in CO₂ linelists was already mentioned.

- line 336-337

Similar values for the changes of the reduced χ^2 for the ATMO and PHOENIX models should be presented to strengthen the claim of the SO₂ detection.

The “Bayesian Gas Removal” technique that led to the SO₂ detection significance of 4.8σ was only developed within the PICASO framework. Similarly, the “Gaussian Residual Fit” technique that led to a detection significance of 3.5σ was developed within the PHOENIX framework. We believe that these two independent analyses for detection significance, along with the additional spectra calculated using the gCMCRT radiative transfer code, are strong enough to support the SO₂ detection.

- line 369

Why was the threshold of 15σ adopted opposed to the default of 4σ ? For the sake of future observers, please give a brief reason. What is the impact when adopting a different threshold?

We tested a number of values for the ramp-jump detection and found that 15-sigma produced the most consistent results at the integration level. The ramp-jump detection is designed to catch cosmic rays in the integration, however, we found that applying cosmic ray detection thresholds in both space and time across the whole sample of integrations more accurately identified them for removal from the data using the sigma clippings described in the next comment.

- line 404-405

Similarly, please give a brief reasoning of the thresholds of 6σ and 20σ , and the impact of different thresholds.

We have clarified the use of these sigma clipping values in the text. 6-sigma is used in the spatial axis to remove permanently deviant pixels. 20-sigma is used in the time axis to remove short-term events such as cosmic rays which are likely to be significantly different to the underlying fluctuations in the data.

- line 430-434

I could not understand the sentences. There are 2 identical sentences here, and it is very confusing. Again, please give a brief reasoning of the thresholds of 5σ and 3σ , and the impact of different thresholds.

The repeating sentences were a typo and the first sentence has been removed.

The 3 sigma threshold was chosen after exploration of both 3 and 5 sigma thresholds. We found that the 5 sigma threshold left remaining bad pixels, so we tested and selected the more conservative 3 sigma threshold.

- line 447

The authors describe "3 standard deviations" here. Is this meaning different from 3σ (e.g., in line 442)? Why is the different notation used here?

This was a typo and should have indeed read 3σ . It has now been corrected.

- line 488-489

Please give a brief reasoning of the thresholds of 10σ and 3σ , and the impact of different thresholds.

The selection of these thresholds was stabilised as standard values for Eureka reductions of JWST data from the initial works with the PRISM and G395H instruments. While the 10-sigma outlier rejection in time has only a small effect on the data and is chosen to only remove clear outliers, we have found that a 3-sigma threshold was a good balance in the background subtraction step, as tests with higher thresholds led to some outlier pixels biasing full columns in the column-by-column background correction, while stricter thresholds were excluding too many pixels and leading to less precise background fits.

The following sentence has been added to this section: “These outlier rejection thresholds were selected to remove clear outliers in the data and provide a balance with the background subtraction step.”

- line 506

Should be "Limb-darkening" according to the following paragraph.

This has now been corrected.

- the section of "Light Curve Fitting"

I highly appreciate the authors' effort for testing those different pipelines and presenting comparisons of resultant transmission spectra. I would further appreciate it if the authors would present useful guidelines for future observers here. Namely, what is the authors' recommendation for future observers based on the authors' experience from the early release science? Do the authors recommend doing a similar analysis (i.e., using 11 pipelines and combined) for future observers? Or is any pipeline fine? Or which pipeline is recommended? Such information would be extremely useful for future observers and worth reporting as a result of the early release science.

We thank the referee for an important point. We have added the following paragraph to the end of the Light curve fitting section before describing each of the independent pipelines.

“While there is a general consensus across each of the data analyses, by comparing the results of each fitting pipeline we were better able to evaluate the impact of different approaches to the data reduction, such as the removal of bad pixels. For future studies, we recommend the application of multiple pipelines that utilise differing analysis methods, such as the treatment of limb-darkening, systematic effects, and noise removal. No single pipeline presented on its own can fully evaluate the measured impact of each effect, given the differing strategies, targets, and potential for chance events such as a mirror tilt with each observation. In particular, attention should be paid to $1/f$ noise removal at the group- versus integration-level for observations with fewer groups per integration than this study.”

- Extended Data Table 1, line 854

The abbreviation of "LD - limb-darkening" is defined here, but not used in the Table. Is this necessary?

This was left over from a previous version of the table and has now been removed as it is not needed.

- the section of "Model Comparison"

The order of 3 models are different in the text (PHOENIX, ATMO, PICASO+Virga) and Extended Data Table 2 (ATMO, PHOENIX, PICASO+Virga).

We have switched the order of the text to match the table.

- Extended Data Table 3, line 924

Although the authors describe "B is the Bayesian evidence," it appears B is the Bayes factor between the models including and excluding a specific opacity source. Is this a typo, or do the authors use the same meaning for the Bayesian evidence and the Bayes factor?

We did mean the Bayes factor, this has been changed.

- line 1004

I understand the 4.8σ detection of SO₂ mentioned in the main text is statistical (Bayesian) significance of the presence of SO₂, and here the authors present the amount of SO₂ based on the Levenberg-Marquardt regression. But apparently the description of SO₂ abundance of 2.5 ± 0.65 ppm does not match 4.8σ detection. To avoid any confusion by readers, please give an explanation on what is the definition of the 1σ uncertainty of 0.65 ppm here.

The SO₂ quote in Extended Data Figure 7 for the best-fit PICASO (2.5 ± 0.65 ppm) does indeed match the 4.8 detection. However, we note there may have been confusion because the PICASO-derived abundance associated with the detection significance of 4.8 is quoted in the text in logarithmic units ($\log \text{SO}_2 = -5.6 \pm 0.1$). Therefore to avoid confusion we have added a parenthetical definition with ppm units as well.

Referee #2 (Remarks to the Editor):

I understand that this paper is one of a series of papers and SO₂ detection is the subject of another paper.

Then I am satisfied with the responses and the revised manuscript